cognition/psychology/neuroscience

interval timing, central tendency effect, ageing, Bayesian observer model, prior distributions

**Author for correspondence:**
Sarah C. Maaß
e-mail: s.c.maass@rug.nl

# Conceptually plausible Bayesian inference in interval timing

Sarah C. Maaß[1,2,3], Joost de Jong[1,2], Leendert van Maanen[4] and Hedderik van Rijn[1,2]

[1]Department of Experimental Psychology, and [2]Behavioral and Cognitive Neurosciences, University of Groningen, Grote Kruisstraat 2/1, 9712TS Groningen, The Netherlands
[3]Aging and Cognition Research Group, German Center for Neurodegenerative Diseases (DZNE), Leipziger Straße 44, 39120 Magdeburg, Germany
[4]Department of Experimental Psychology, Utrecht University, Heidelberglaan 1, 3584 CS Utrecht, The Netherlands

 SCM, 0000-0002-8122-6604; JdJ, 0000-0001-8841-5646; LvM, 0000-0001-9120-1075; HvR, 0000-0002-0461-9850

In a world that is uncertain and noisy, perception makes use of optimization procedures that rely on the statistical properties of previous experiences. A well-known example of this phenomenon is the central tendency effect observed in many psychophysical modalities. For example, in interval timing tasks, previous experiences influence the current percept, pulling behavioural responses towards the mean. In Bayesian observer models, these previous experiences are typically modelled by unimodal statistical distributions, referred to as the prior. Here, we critically assess the validity of the assumptions underlying these models and propose a model that allows for more flexible, yet conceptually more plausible, modelling of empirical distributions. By representing previous experiences as a mixture of lognormal distributions, this model can be parametrized to mimic different unimodal distributions and thus extends previous instantiations of Bayesian observer models. We fit the mixture lognormal model to published interval timing data of healthy young adults and a clinical population of aged mild cognitive impairment patients and age-matched controls, and demonstrate that this model better explains behavioural data and provides new insights into the mechanisms that underlie the behaviour of a memory-affected clinical population.

## 1. Introduction

In a world that is uncertain and noisy, perception makes use of optimization procedures to reduce the influence of moment-to-moment noise by incorporating statistical properties of previous

experiences. This observation holds for the perception of many psychophysical quantities [1], including the estimation of distance [2] and angles [3], object size [4] and duration [5,6]. These types of optimization procedures assume that when a specific stimulus needs to be reproduced, observers do not only take the current percept into account but also incorporate their prior knowledge of previous similar incidents to form an internal estimate of this stimulus. This process yields more optimal average responses when the perception of quantities is noisy, with the central tendency effect [4,7] as its prime signature. Hollingworth's work focused on 'time sense', describing that reproductions of durations gravitate towards the mean, with durations above the mean being underestimated and durations below the mean overestimated. Later work has linked this central tendency effect with the scalar property: as the noise increases with the size of an interval [8], the prior experiences will have a relatively larger influence on the longer percepts, and one would, therefore, expect a stronger central tendency bias for longer durations. Even though the central tendency effect was one of the first timing phenomena described in the literature, a formal account of this phenomenon has only recently been proposed.

In 2010, Jazayeri & Shadlen [5] described how Bayesian principles can be used to construct an elegant mathematical framework in which an observer is assumed to reproduce a duration by integrating the perceived duration, represented as a distribution that can vary in noisiness (the likelihood distribution), with a probability distribution (the prior) representing the earlier observed durations. The multiplication of prior and likelihood results in the posterior distribution of which the mean is taken as the internal estimate of the to-be-reproduced duration (figure 1, left column). The observed reproduced duration is a function of this estimation and a noise component reflecting various noise sources. In Jazayeri & Shadlen's work, the prior is conceptualized as a uniform distribution defined over the stimulus range and effectively acts as a filter constraining the likelihood to the presented range of durations. To reflect the increased temporal uncertainty for longer intervals, the model assumes a linear scaling of the width of the likelihood as a function of the perceived duration. This functional implementation of the scalar property [8] allows the model to account for the empirical observation that the strength of the central tendency bias is a function of the magnitude of the duration. Similarly, this model accounts for individual differences in central tendency magnitudes by assuming differences in the variability of the temporal percept: the noisier the likelihood, the stronger the impact of the prior, and thus the stronger the central tendency effect (see dark and light lines in the likelihood and posterior panels of figure 1, and https://vanrijn.shinyapps.io/MaassVanMaanenVanRijn-Fig1 for a dynamic simulation demonstrating these effects). The second source of individual differences in Jazayeri & Shadlen's model is the 'motor-noise' distribution that was estimated per participant to reflect the noise associated with the mapping from the mean of the posterior distribution to the actual reproduced duration.

Jazayeri & Shadlen's model is a specific instance of a family of Bayesian observer models. These models are characterized by a mapping between stimulus and response, in which the likelihood of an observed stimulus is weighted by a prior distribution over stimuli. Different distributions or functions could be hypothesized for the likelihood, the motor-noise components and the prior. Indeed, over the last couple of years, various Bayesian observer models have been shown to accurately reproduce human behaviour in a number of timing tasks (for a review, see [9–14]). Most notably, Cicchini *et al*. [9] proposed an alternative Bayesian observer model in which the prior is represented by a Gaussian distribution of which the width varies as a function of the central tendency effect. This allows the model to capture the magnitude of the central tendency effect partially using a memory-based explanation, instead of solely relying on variability in the likelihood component (see the middle column of figure 1). Even though the suitability of this model to explain empirical data has been questioned (see [15]), both Cicchini *et al*.'s work and Narain *et al*.'s evaluation does fit with the assumption that the choice of the prior distribution is an integral part of Bayesian modelling and should be driven by theoretical and empirical considerations (cf. [16–19]).

In the current paper, we will evaluate the Bayesian observer models proposed in the timing field in terms of their construct validity and assess how the models can explain timing performance in different populations. In this context, construct validity refers to the question whether the computational models accurately reflect the constructs that account for variance in observed performance, similar to the construct validity of a psychological test (see, e.g. [20]). Based on these evaluations, we will argue that the assumed prior distributions are, though mathematically elegant, too simple to account for performance in non-idealized laboratory settings, that specific properties of the underlying distributions do not translate into realistic predictions, and that certain assumptions related to perception and reproduction noise need to be re-evaluated. Based on model simulations and comparisons with earlier published datasets of healthy young adults [21] and an aged sample [22]

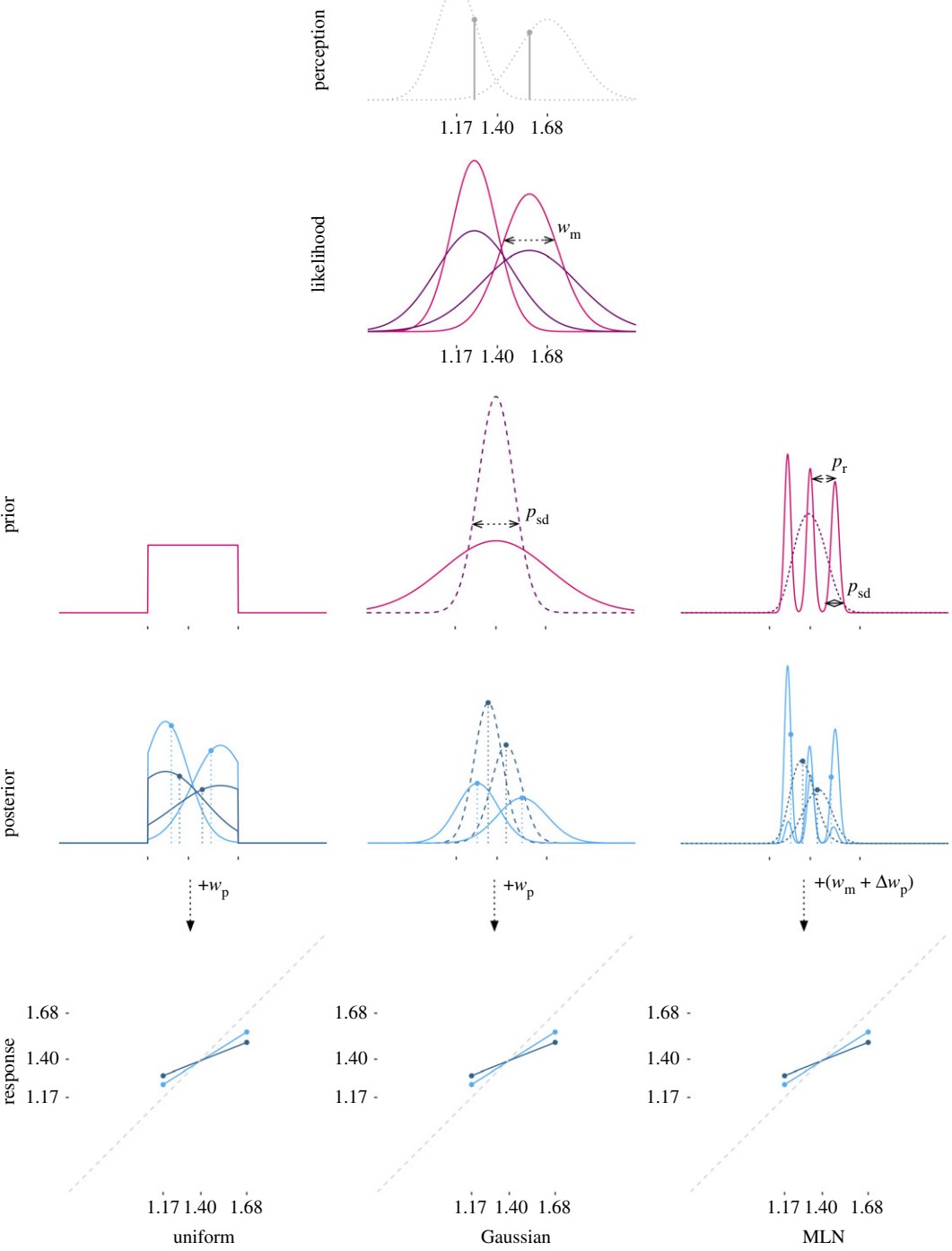

**Figure 1.** A schematic representation of Bayesian inference in interval timing. The left-most column depicts, from top to bottom, how the width of the likelihood determines the magnitude of the central tendency effect, and the middle and right columns depict how a similar effect can be obtained manipulating the prior. In all columns, the top figure depicts the likelihoods associated with perceived short or long durations. The second row depicts the prior, either a uniform prior (left, cf. [5]), a Gaussian prior (middle, cf. [9]) or the mixture lognormal (MLN) prior. The third row depicts the posterior distribution with the dots reflecting the means of the distributions, which is the estimate (this figure does not depict the influence of the motor noise). The fourth row shows the resulting central tendency effects. In the left column, the light blue lines reflect the posterior resulting from a rather narrow likelihood distribution, representing a fairly accurate internal clock. The dark blue lines represent a noisier clock, resulting in a stronger central tendency effect. The posterior and response panels in the middle and right column only show the narrower likelihood distribution for legibility. In the middle plot, the pink, solid line represents a wider, and thus less informative prior, resulting in a relatively small adjustment of the likelihood. The purple, dashed line represents a more constrained prior, resulting in larger adjustments, and thus a larger central tendency effect. In the right plot, the pink, solid line represents a mixture in which three narrow distributions are distributed over most of the stimulus range, whereas the dashed, purple line represents the mixture of three wider distributions that are maximally pulled towards the mean. All mixture components are equally weighted.

that consists of patients diagnosed with mild cognitive impairment (MCI, a clinical diagnosis considered as precursor to Alzheimer's disease and dementia, [23]) and healthy, aged controls, we will argue that a prior consisting of a mixture of lognormal distributions is preferred, and that the motor noise associated with the reproduction of a duration needs to be estimated separately from the noise associated with the timing process itself.

## 1.1. First theoretical consideration: shape of the prior

The elegance of the specific Bayesian observer model proposed by Jazayeri & Shadlen [5] is partially due to a number of simplifying assumptions. For example, the prior is assumed to be a uniform distribution spanning the range of the presented durations. Even though this provides computational simplicity, its theoretical suitability can be questioned as the nature of the central tendency effect is that extreme observations are pulled inwards. Thus, if one assumes that the prior is constructed on the basis of previous posterior distributions (following both Bayesian principles and instance-based explanations of the central tendency effect, e.g. [24,25]; for reviews, see [10,26]), the resulting prior must have a higher mass at the centre than at the more extreme values.

One step towards a more plausible prior was explored by Cicchini *et al.* [9] who proposed a Gaussian prior distribution. Even though this distribution has characteristics that are theoretically more plausible than those of a uniformly distributed prior, its theoretical elegance is affected by the necessity to constrain the range of this Gaussian prior to prevent it extending to negative values. Moreover, even though the heavier centre of the Gaussian captures the bias towards the mean, the repeated presentation of different durations that are all reasonably accurately reproduced should result in a prior distribution that is probably better characterized as a mixture of distributions centred at the presented durations. This rationale is supported by the observations of Acerbi *et al.* [12] who have demonstrated that participants can acquire an internal approximation of more complex empirical distributions that notably deviate from either Gaussian or uniform distributions as multiple peaks can be observed in the approximated prior distributions (figure 5 and the corresponding section in the 'Results and discussion' for a more elaborative review).

Thus, both theoretical considerations and empirical data suggest that the prior of temporal reproduction yields a mixture distribution. Even though a mixture of Gaussian distributions could be explored, this would require a decision about how to constrain the lower tail of these distributions. Moreover, empirical work has demonstrated that response functions in timing studies are better described by skewed distributions such as the Wald or inverse Gaussian, skewed-normal or lognormal distribution (see, e.g. [27–29], and the discussion in [9]; for a mathematical approach, see [30]). Given these considerations, a theoretically reasonable prior would consist of a mixture of skewed distributions each representing a stimulus duration. Here, we explore a mixture-based prior that combines a lognormal distribution for each unique duration that was presented to the participants. We will refer to this model as the mixture lognormal (or MLN) model.

A variety of empirical and theoretical justifications have been proposed for skewed distributions of timed responses [31]. In the MLN model, we use a mixture of lognormal distributions as the prior for time intervals. Put differently, the prior represents logarithmically transformed time intervals on a linear scale. Although it is not straightforward to empirically dissociate linear versus logarithmic encoding of time, some psychophysical [32–34] and neurophysiological evidence [35] suggests a logarithmic encoding of time. Moreover, several Bayesian models of time estimation have assumed that Bayesian integration takes place on a logarithmic scale [36–38], which naturally accounts for lognormally distributed time estimates and the scalar property. The MLN model departs from these logarithm-based Bayesian models in two theoretically important ways: (i) assumptions about the scalar property and (ii) the scale at which Bayesian integration is applied: logarithmic or linear.

An important reason for using a logarithmic representation of time is that it naturally produces the scalar property without additional *a priori* assumptions. The MLN model, in contrast, assumes *a priori* the scalar property in the likelihood function. However, it is difficult to dissociate empirically which assumption should be considered more fundamental. Assuming logarithmic encoding of some quantity naturally produces the scalar property [3]; conversely, assuming the scalar property on a linear scale in combination with Bayesian integration on this linear scale naturally produces a logarithmic compression of responses [39]. It is an interesting question which phenomenon (if any) is more fundamental, but the MLN model is not equipped to settle this theoretical issue.

Previous logarithm-based Bayesian models have assumed that Bayesian integration already takes place on a logarithmic scale and is only transformed to a linear scale when it produces final

(lognormally distributed) outputs. The MLN assumes that, after an interval is encoded logarithmically, Bayesian integration happens on a linear scale. The crucial difference, therefore, is whether Bayesian integration happens at an early logarithmic scale, or at a late linear scale. This issue cannot be settled by the experiments we model in this paper. However, some empirical evidence suggests that Bayesian integration in temporal reproduction takes place at late rather than early stages. When participants were required to produce intervals that were a proportion of the sampled intervals, their pattern of bias and variance suggested that Bayesian integration took place after they transformed the perceived interval (i.e. multiplied it with a scaling factor) for later temporal production [40]. Therefore, if intervals are first transformed to a logarithmic scale and then to a linear scale, Bayesian integration may well happen on a linear scale, as the MLN assumes.

The shape of the compound mixture distribution is defined by three parameters that can be directly mapped onto constructs. Two parameters are linked to the individual distributions: the location of the mean of the distribution and the distribution's standard deviation. With respect to the means, we will assume that the location of the middle mixture component corresponds with the duration of the middle stimulus, and that the locations of the other mixture components reflect the central tendency effect, with more extreme distributions pulled inwards as a function of the magnitude of the central tendency effect. This could be a linear shift, but one could also assume a more flexible shift with longer durations being more affected than shorter durations. Here, we will assume that the means of the distributions will be distributed as a geometric series, matching the distribution of the presented stimulus durations. The standard deviation of these components will scale linearly with the estimated means, following the scalar property. We will estimate the standard deviation for the component associated with the median stimulus duration and derive the standard deviations for the other components. See the right column of figure 1 for an illustration. The third parameter that could influence the shape of the overall compound mixture is the weight of each of the individual components. Even though this parameter could be used to capture non-uniform stimulus distributions, we will not explore this additional source of variability as the modelled datasets all presented uniform stimulus distributions (also see the section 'Fitting empirical prior distributions' in the 'Results and discussion' section).

## 1.2. Second theoretical consideration: sources of noise

A second consideration associated with the Bayesian observer model is that it incorporates two sources of noise, one associated with the perceptual processing of the presented duration ($w_m$), determining the width of the likelihood, and one associated with the mapping of the posterior to the actual reproduced duration ($w_p$). In other domains, $w_m$ and $w_p$ are typically conceptualized as representing perceptual noise and non-specific, motor or response noise [2,3]. Importantly, the nature of interval timing tasks adds a noise component in that the perception of duration is not momentaneous. Instead, the brain needs to read out a noisy timing system to perceive the duration of an interval, and the same or a similar timing system is involved during reproduction as the timing system needs to generate a cue upon which the motor response can be executed. This suggests that both $w_m$ and $w_p$ should incorporate a 'clock-noise' component, in addition to the more standard perceptual and motor-noise components. Thus, $w_m$ captures the perceptual noise associated with perceiving the onset and offset of the presented duration and clock noise associated with the actual timing of the interval, where $w_p$ captures the perceptual noise for the onset of the reproduction phase, the clock noise and the motor noise associated with the motor movement to mark the end of the reproduction phase. As the scalar property is reflected in noisier clock estimates with increased durations, the noise associated with both perception and production stages should scale with the relevant duration. As motor noise is more variable and larger than visual perception noise [41], $w_m$ should be smaller than $w_p$ when these parameters are independently estimated as the production stage entails the combination of perceptual and clock noise (i.e. $w_m$) and the motor noise associated with the execution of the motor movement.

As both parameters were fit independently in Jazayeri & Shadlen's Bayesian observer model [5], $w_p$ could take smaller values than $w_m$ and no correlation between both parameters was enforced. In Cicchini *et al.*'s work [9], neither of the noise parameters were estimated. Their model incorporated an empirical estimate for $w_m$ that was based on each participant's performance on a bisection task, and $w_p$ was fixed for all participants. Similar to Jazayeri & Shadlen's model, this model did not enforce that $w_m$ should be smaller than $w_p$, nor that they are correlated. Additionally, by keeping $w_p$ fixed over all participants, Cicchini *et al.* made the simplifying assumption that all sources of noise, including clock noise, are identical for all participants during reproduction. Moreover, as the reproduction noise was not

influenced by the estimated duration, Cicchini's model implicitly assumes a timing system that does not adhere to the scalar property during reproduction. However, in the discussion of their model, Cicchini *et al.* already indicate that the noise should probably be estimated on a per-participant basis. Thus, following Cicchini *et al.*'s recommendations and Jazayeri & Shadlen's implementation, we do individually estimate the Weber fractions of both clock and motor noise. To provide a fair comparison between models, we extended our implementation of Cicchini *et al.*'s model to allow for individual differences in both perception and reproduction noise. As Cicchini *et al.* [9] estimated reproduction noise at 0.1 (referred to as motor noise), and allowed the clock noise to take on values higher than 0.1, we did not constrain the estimation procedure with respect to the relative values of $w_p$ and $w_m$ for our implementation of their model.

For the MLN model, we—similarly to Jazayeri & Shadlen and Cicchini *et al.*—scaled the standard deviations of the individual components by their means. Based on the theoretical considerations outlined above, we defined $w_p$ as $w_m + \Delta w_p$, where $\Delta w_p$ reflects the noise associated with the motor processes. Following earlier work, we assume the production noise to follow a Gaussian distribution, which results in $N(t_s, (w_m \times t_s))$ for the perception stage as clock noise is scaled by the presented duration, and $N(t_e, (w_m \times t_e) + \Delta w_p)$ for the reproduction stage.

## 1.3. Summary and overview of the paper

In the following sections, we will demonstrate how the MLN model, including its theoretical constraints on the noise distribution, compares with two baseline models published in the literature. These three Bayesian observer models take into account individual differences by assuming variations in clock (i.e. likelihood width) and motor noise, and, for two of the three models, allow for variations in the memory representation of previously perceived durations (i.e. width and/or shape of the prior). The first baseline model is an implementation of the model described in Jazayeri & Shadlen [5] that assumes a uniform prior constrained over the range of presented durations and allows for an unconstrained estimation of $w_m$ and $w_p$. We will refer to this model, after the shape of its prior, as the Uniform model. The second baseline model is our implementation of Cicchini *et al.*'s [9] model that assumes a Gaussian prior. Unlike the model reported in Cicchini *et al.* in which production noise was constant over participants and the perception noise was derived from a separate experiment, our implementation provides more freedom and allows for individual differences in both clock and reproduction noise. For all three models, we will integrate the likelihood and prior numerically over a range extending to 0.5 and 1.5 times the minimum and maximum stimulus durations. In addition, we will estimate a single parameter representing the prior's standard deviation for the Gaussian and MLN model. As the central tendency effect assumes that the extreme distributions are pulled towards the mean, we estimate an additional parameter representing the inward pull for the means of the outer distributions in the MLN model. This factor allows for the mean of the distributions of the prior to be located either at the original locations of the empirical distribution (resembling a wide prior, reflecting no central tendency effect), or, in the other extreme case, to all be identical to the median value of the empirical distribution (resembling a more narrow prior, reflecting a maximal central tendency effect).

The Uniform, Gaussian and MLN models will be fit to data from a sample consisting of 63 young healthy adults (undergraduate students) who performed a 1 s estimation task that was used to assess clock variability, and a multi-duration reproduction task ([21]; figure 2). The multi-duration reproduction task consisted of reproduction of three equiprobable intervals (1.17, 1.4 and 1.68 s). The three presented durations are defined as 1.4/1.2, 1.4, 1.4 × 1.2, following a geometric series with scale factor 1.4, common ratio 1.2, and calculated for terms −1, 0 and 1. As expected, the data demonstrates the central tendency effect, shown in figure 2a.

Furthermore, to test the hypothesis that a mixture model can provide new meaningful interpretations at group levels, the MLN model was fit to the data of 10 MCI individuals and 25 age-matched healthy controls (HC) from Maaß *et al.* [22] (figure 2b). These participants performed the same 1 s estimation task that was used to assess clock variability and a multi-duration reproduction task as the students' sample in Maaß & Van Rijn [21]. Interestingly, the memory-impaired population showed a stronger central tendency effect than the HC. The observed negative correlation between the magnitude of the central tendency effect and memory functioning might be considered paradoxical as it implies that with decreasing memory functioning, the prior—which can be seen as the memory for previously perceived stimuli—has stronger influence on subsequent performance (see [22] for a discussion). Even though differences in clock variance, resulting in a wider likelihood for MCI participants, could be a

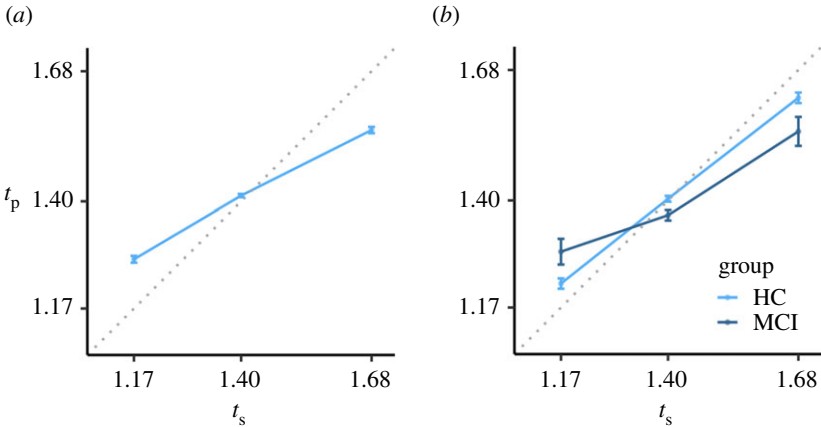

**Figure 2.** Empirical results of the multi-duration reproduction task as reported in Maaß & Van Rijn [21] (*a*), and in Maaß *et al*. [22] (*b*). (*a*) represents the data of 63 healthy young adults; (*b*) represents the data of 10 mild cognitive impaired patients (MCI) and 25 age-matched, healthy controls (HC). Error bars are standard errors of the mean with the within-participants Cousineau–Morey correction applied. Data in both panels adhere to the central tendency effect, with a larger central tendency effect of the MCI patients than for the HC in (*b*). $t_s$: presented interval; $t_p$: produced interval in seconds.

possible explanation, the temporal production variability as assessed by the 1 s estimation task was similar among all groups. Thus, the results suggest that MCI patients have a stronger influence of prior experiences than age-matched HC resulting in stronger central tendency effects. The MLN model that we present here will quantify these counterintuitive effects by disentangling memory and clock variability processes.

# 2. Material and methods

## 2.1. The Bayesian observer model

We fit Bayesian observer models to the data from the experiments introduced above. All models follow the rationale from Jazayeri & Shadlen [5] that a reproduced temporal interval on a particular trial depends on (i) the relationship between the sample interval $t_s$ and the internal measurement of this interval ($t_m$), (ii) the mapping of $t_m$ onto an internal representation of the perceived duration ($t_e$) by incorporating the prior, and (iii) the incorporation of noise to map the internal representation to the produced interval ($t_p$). The likelihood of a specific internal representation of the sample interval is modelled as

$$p(t_s) = \varphi\left(\frac{t_s - t_m}{w_m t_s}\right)$$

and

$$p(t_m | t_s) = \varphi\left(\frac{t_s - t_m}{w_m t_s}\right).$$

Here $\varphi(\cdot)$ represents the standard normal distribution function, and $w_m$ is the Weber fraction associated with the internal representation of the sample interval (i.e. perception clock noise). The scalar property is captured by the multiplication of $w_m$ and $t_s$, as this results in increasing uncertainty with longer durations. Likewise, the likelihood of a specific reproduced duration is modelled as

$$p(t_e) = \varphi\left(\frac{t_e - t_p}{w_p t_e}\right)$$

Here, $w_p$ is the Weber fraction associated with the reproduction, providing another locus for the scalar property. The mapping from the internal measurement ($t_m$) to the internal estimate ($t_e$) depends on the integration of the likelihood over a prior distribution. The specification of the prior distribution will differ according to the different theoretical considerations discussed above, resulting in three distinct models.

First, the *Uniform model* is a direct implementation of the Jazayeri & Shadlen [5] model that assumed a uniform distribution of the prior. The range of the uniform distribution is determined by the stimulus range (i.e. ranging from 1.17 to 1.68 s in the experiments we report on here). Second, the *Gaussian model* assumes a normally distributed prior (cf. [9]). The prior is centred at the median stimulus duration (i.e. 1.4 s), and truncated below 0. The parameter that is estimated is the standard deviation ($p_{sd}$) of the normally distributed prior. Third, the *MLN model* incorporates the theoretical considerations that were previously introduced. Specifically, the MLN model is characterized by a mixture of multiple lognormal distributions. We assume that the number of mixture components equals the number of empirically presented durations. The standard deviation of the mixture components is estimated under the constraint that they scale linearly with the component means (following the scalar property). Consequently, below we report the standard deviation ($p_{sd}$) of the middle component and the common ratio $p_r$ of the geometric series that scales the mixture components assuming a scale factor of 1.4 and terms −1, 0 and 1. Finally, we assume that the weight of each mixture component is equal, representing the fact that participants have observed every stimulus an equal number of times.

For all three models, we additionally estimate the Weber fraction parameters $w_m$ and $w_p$. For the Uniform and Gaussian models, both $w_m$ and $w_p$ are unconstrained, conforming to the specifications in Jazayeri & Shadlen [5] and Cicchini *et al.* [9]. However, the MLN model incorporates our second theoretical consideration that $w_p$ should reflect the combination of clock noise captured by $w_m$ and additive motor noise. We, therefore, estimate $\Delta w_p$ and define $w_p$ as $w_m + \Delta w_p$.

## 2.2. Model fitting and model selection

For each participant, we performed a linear grid search for the optimal set of parameters [42–44]. The range as well as the resolution of each parameter is presented in table 1. The ranges are based on an iterative process to ensure an optimal resolution and to prevent ceiling and floor effects. The number of values tested in the range was chosen to provide a fine enough grid, while still being able to search the full grid in a reasonable amount of time. For each parameter vector, we generated 1000 trials for each stimulus duration and computed the RMSE between the observed and predicted 10, 30, 50, 70 and 90% quantiles separately for every participant. As we are mainly interested in the central tendency effects, and not in any linear shifts that represent structural over- or under-reproduction, we subtracted, for each reproduction, the average reproduction time across all trials and added the average stimulus duration (following [9]). The integration of the likelihood over the prior distribution was approximated using the Riemann sum/rectangle method over the range 0.585–2.52 s, to avoid misspecification where the posterior density approaches zero (implementation details of the models, grid search data, code for all figures, and fitting routines can be found online: https://osf.io/kqjxf/, [45]).

It is common practice to compare statistical models in terms of their quantitative fit to the data, taking into account any differences in model complexity. Here, we take a different approach, as our research question is not about the model that provides the best balance between goodness-of-fit and model complexity, but about which model provides the best understanding of the empirical phenomena observed in healthy and clinical populations while adhering to plausible theoretical considerations. For that reason, we will consider the validity of the estimated parameters vis-a-vis their interpretation to differentiate between models (see [46,47], for similar discussions).

When comparing model fits to the data of individual participants, we will plot the empirical data and the model estimates for the vector of parameters with the lowest RMSE to reflect the best possible fit we could obtain. However, this depiction of the model fit ignores the uncertainty in model selection: it is well possible that another vector of parameters provides similar fits in terms of RMSE. Therefore, when interpreting the parameters, we did not use the value with the lowest RMSE, but we calculated a weighted average of the parameters using Bayesian model averaging (weighted by their respective model probability using relative Akaike information criterion (AIC) values) that best reflects the information provided by the model fitting processes [48,49].

## 2.3. Bayes factors

All inferential analyses on the model parameters are based on Bayes factors that were computed with the R package BayesFactor (v. 0.9.12-4.2; [50]) using the default prior settings and are interpreted based on the guidelines provided by Jeffreys [51], see also [52]). The reported Bayes factors summarize the extent to

**Table 1.** Parameter ranges for the estimated parameters (with the number of tested equidistant values listed in parenthesis).

| model | $w_m$ | $w_p$ | $\Delta w_p$ | $p_{sd}$ | $p_r$ |
|---|---|---|---|---|---|
| Uniform | [0.005–0.2] (20) | [0.005–0.2] (20) | | | |
| Gaussian | [0.005–0.2] (20) | [0.005–0.2] (20) | | [0.01–0.3] (20) | |
| MLN | [0.005–0.2] (20) | | [0.0001–0.15] (20) | [0.01–0.3] (20) | [1.02–1.2] (10) |

which an observer's opinion of the tested variable should change based on the data; in our case, whether the parameters differ between groups of participants. A Bayes factor of 1 indicates that both hypotheses are equally likely under the data and therefore is inconclusive. Bayes factors larger than 1 represent evidence for the alternative hypothesis of an influence of the tested independent variable on the dependent variable (i.e. the groups differ with respect to the parameter of interest), and Bayes factors less than 1 represent evidence for the null hypothesis of no effect of the tested variable (i.e. the groups do not differ).

# 3. Results and discussion

## 3.1. Fitting the student sample

First, we will fit the three models to the data of 63 healthy young adults and determine the best fitting vector of parameters per participant and model. Figure 3 depicts in the three left-most columns the empirical data and the model fits resulting in the lowest RMSE for three participants. In addition, we plotted in the right-most column of figure 3 the predicted reproduced duration against the empirical reproduced duration, for all participants for each quantile (coded by colour) and empirical duration (each diagonal line representing a $t_s$). At first sight, all models seem to provide an equally good fit, as no strong deviations can be observed from the diagonal. Yet, when comparing the squared difference between predicted and empirical duration per type of model (RMSEs: 0.032 for Uniform, 0.028 for Gaussian, 0.027 for MLN), there is strong evidence (BF = 11.31) that this squared deviation is better predicted when type of model is included as main effect (in addition to the main and interaction effects of quantile, $t_s$, and empirical duration). This effect is driven by a worse fit for the Uniform model, as there the Gaussian and MLN models provide comparable fits (BF = 0.055). Thus, although all three models provide a reasonable visual fit to the data, the Gaussian and MLN models better account for the data than the Uniform model.

As discussed in the Introduction, the Uniform and Gaussian models do not constrain the estimation of the width of the noise of the reproduction ($w_p$) in relation to the width of the noise of the perception ($w_m$) of the duration. As a reproduction consists of the same clock processes associated with the perceptual stage plus an additional motor response, one should expect $w_p$ to be larger than $w_m$. Figure 4 depicts the difference between $w_m$ and $w_p$, expressed as $\Delta w_p$, for the Uniform and Gaussian models, and the estimated $\Delta w_p$ for the MLN model. As the two left-most violin plots show that a sizable proportion of the best fitting Uniform and Gaussian models assume a *larger* $w_m$ than $w_p$, this suggests that for those participants the reproduction of a duration is associated with less noise than the perception of a duration. It is important to note that this theoretically implausible result is intrinsic to the Uniform model as the only way this model can account for the magnitude of the central tendency effect is by manipulation $w_m$. As the prior is fixed, $w_m$ captures the central tendency effect: a very small $w_m$ yields a very peaked likelihood which mostly falls within the defined uniform prior, and thus only small central tendency effects will be observed, whereas a large $w_m$ will result in large proportions of the likelihood falling outside the range of the prior, resulting in stronger pull effects. Thus, for participants with large central tendency effects, $w_m$ needs to be large. If $w_p$ would be constrained to be larger than $w_m$, this would result in too dispersed estimates for the extreme quantiles, and thus $w_p$ will often be smaller than $w_m$ when a participant shows a large central tendency effect. Note that for the Gaussian model, constraining $w_p$ might be an option as central tendency effects might also be captured by manipulating the width of the prior; however, we opted for keeping the models as similar to possible to the earlier published descriptions. This results in two models with parameter estimates that, even though they provide reasonable fits to the data, are

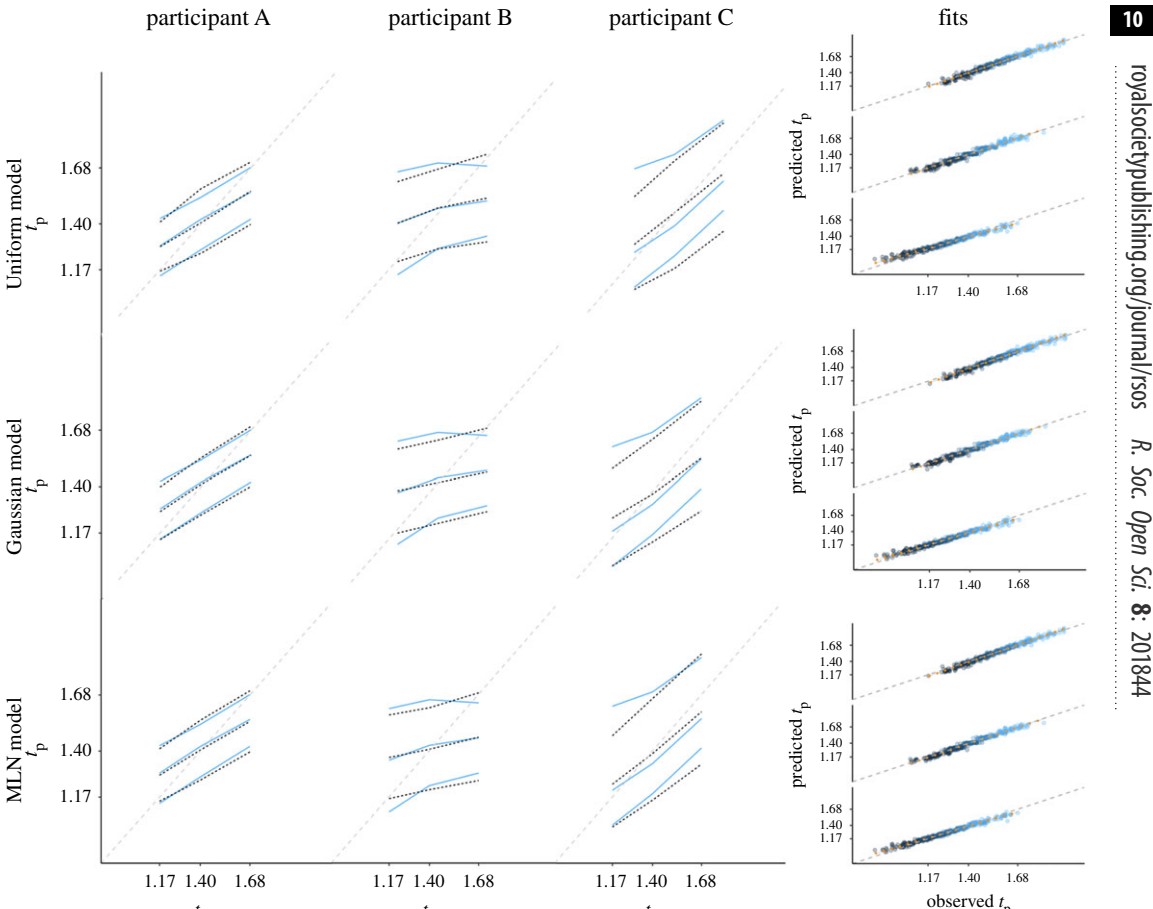

**Figure 3.** Fit of the models to the empirical data from Maaß & Van Rijn [21]. The three left-most columns depict the 10, 50 and 90% quantile (in blue) and model fits (in black), for three example participants ($t_s$: presented interval; $t_p$: produced interval in seconds). The three rows represent the Uniform, Gaussian and MLN model, respectively. The right-most panel depicts, for each of the three $t_s$ durations (one diagonal line per $t_s$ duration, longest at top), the relationship between fitted quantiles and the observed empirical quantiles for all 63 participants, with colour scale representing the 10 (black), 30, 50, 70 and 90% (blue) quantiles.

difficult to align with the theoretical constructs they represent. For the MLN model, shown in the right-most violin plot, $w_p$ is defined to be larger than $w_m$, as this model estimates $\Delta w_p$ as a positive value that is added to $w_m$ to arrive at $w_p$.

## 3.2. Fitting empirical prior distributions

A distinction between the Uniform and Gaussian models, on the one hand, and the MLN model, on the other, is that the latter can accommodate more irregular or complex prior distributions. These distributions could either be elicited by stimuli sampled from non-uniform distributions, or because of observing behaviour that does not align with the assumption of an ideal Gaussian or uniform prior. To assess how well the MLN model fits such prior distributions, we fitted this model to the non-parametrically estimated prior distributions of Experiment 1 reported by Acerbi et al. [12] that are shown as the dashed lines in figure 5. In this experiment, participants were asked to perform a multi-duration reproduction task in which the durations are sampled from uniform, bimodal or skewed distributions. Acerbi et al. demonstrated that reconstructions of the subjective priors from the empirical data are roughly in line with the empirical distributions, even though some higher-order statistical features (e.g. multi-modality) are less accurately covered. Here, we will focus on the biased and the uniform distributions tested by Acerbi et al. In the top-left and both bottom panels (figure 5a, c and d), the estimated priors resulting from the presentation of a short (purple/left) and a long

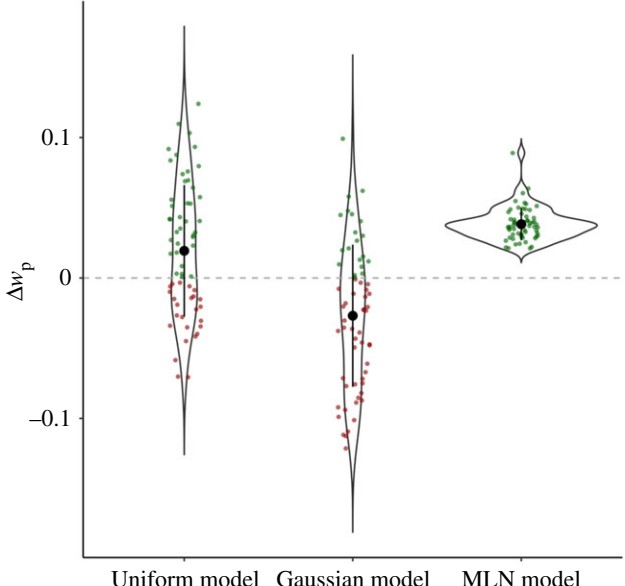

**Figure 4.** Estimated additional noise ($\Delta w_p$) associated with the reproduction of a duration compared with the perception of a duration ($w_p$) for the three models (whisker plot represents mean and two standard deviations). For the Uniform and Gaussian models, $\Delta w_p$ is calculated by subtracting the estimated $w_m$ from the estimated $w_p$; for the MLN model, $\Delta w_p$ is estimated as additive noise to $w_m$ when the estimated duration is mapped to the reproduced duration. Green dots represent participants for whom the estimated combination of clock and motor noise is *larger* than just clock noise, and red dots represent participants for whom the models estimated lower noise for reproduction than for perception.

(pink/right) uniform stimulus distribution of six stimulus durations are shown. These priors are characterized by a central peak flanked by two lower peaks or shoulders, suggesting (i) that the central tendency effect resulted not just in a behavioural pull, but that this pull is also reflected in the central peak of the prior itself and (ii) that the more extreme stimulus durations might be encoded separately from a representation of the mean of the distribution, resulting in the two subordinate peaks. The dashed lines in the top-right panel (*b*) represent both a biased (left/green) and a uniform distribution (right/purple), tested over the same range of stimulus durations. For the biased distribution, the second shortest stimulus had a 7/12 probability of being presented, while the other five stimuli durations were each presented with a probability of 1/12. The relative frequency with which the stimuli durations are presented is shown in the small vertical bars just above the *x*-axis. The estimated prior distributions clearly show multiple modes, again suggesting that a mixture distribution would best describe these priors, with the biased distribution shifted in the direction of the positively biased duration. The bottom two panels (*c* and *d*) depict the two uniform empirical distributions with slightly different parametrized MLN priors, explained in more detail below.

The partly transparent lines plotted in the four panels of figure 5 represent 10 instances of MLN priors. Each prior consists of six component distributions, one for each stimulus duration presented, with a weight between 1/15 and 10/15. The weights, locations and a single standard deviation were estimated by minimizing the summed squared differences between the non-parametrically estimated prior distributions and the MLN prior distribution over the 200 to 1600 ms range (using the Simplex procedure, [53]). In the top row, the standard deviation and both location and weight of each of the components are independently estimated. In the bottom row, the left panel (*c*) depicts the fit assuming that the weights of each component are equal, whereas the right panel (*d*) depicts the fit assuming that the locations of the components equate the presented $t_s$ values. The vertical bars extending from 0 downwards represent the location (*x* value, jittered for visualization purposes) and weight (length of line) of each of the components of the mixture. The small lines at the bottom of each panel represent the relative proportion and location of the presented, empirical durations. By plotting 10 fitted models, the partly transparent lines provide a visual indication of the variability and flexibility of the model fits.

Figure 5*a* demonstrates that the central peak in the estimated prior distribution is captured by a clustering of relatively heavily weighted component distributions around the mean of the empirical

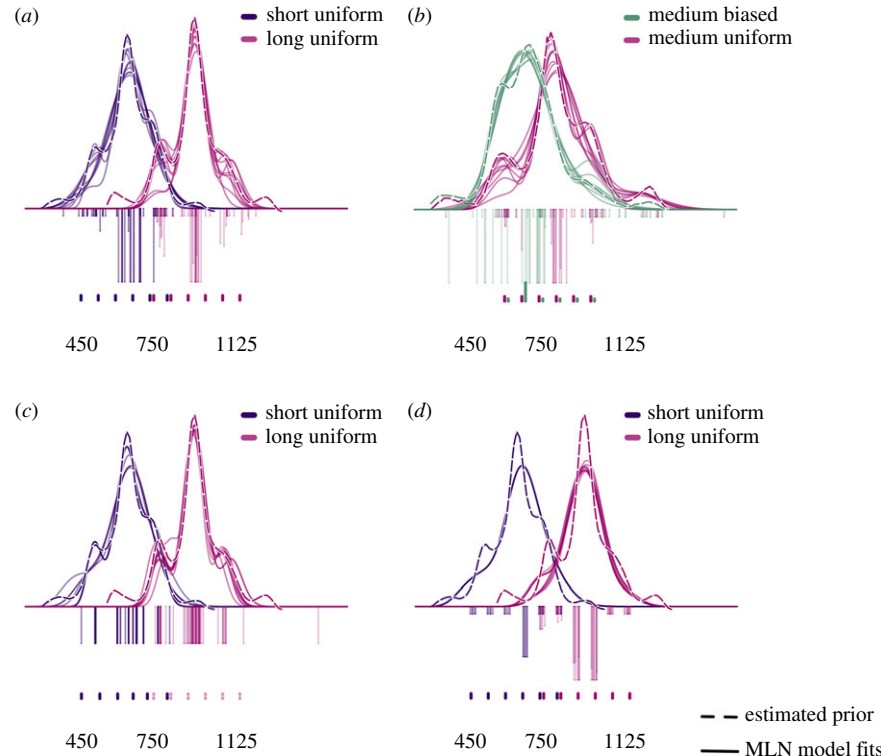

**Figure 5.** The non-parametrically estimated prior distributions of Experiments 1 (panels *a*, *c* and *d*) and 2 (panel *b*) of Acerbi *et al*. [12] and the fits of 10 runs of the MLN model. For the fits in the top two panels, location and weight of each of the components was independently estimated, in the bottom two panels either weight (*c*) or location (*d*) was kept constant. See text for further explanation.

distribution, with the subordinate peaks captured by two smaller clusters to the sides. In figure 5*b*, the downwards pointing location and weight indicators show that the central cluster of the biased condition plotted in green is slightly more scattered and shifted leftwards. These observations demonstrate that the MLN prior can fit the qualitative patterns of the non-parametrically estimated priors reported by Acerbi *et al*. [12]. To assess whether both locations and weights need to be estimated to provide a reasonable fit, figure 5*c* and *d* represent model fits in which either the locations are estimated but the weights kept constant (figure 5*c*), or vice versa, the weights were estimated but the locations kept constant. As can be seen by comparing both panels, figure 5*c* fares better in capturing the kurtosis and multi-modality of the estimated prior. Figure 5*c* can be interpreted as a shift of the mean or location of the prior component that is associated with a particular stimulus duration, a view that fits with the theoretical assumption that the prior reflects the history of posterior values which are also pulled towards the centre of the distribution. To conclude, the MLN model is capable of capturing realistic, more complex prior distributions. When comparing figure 5*c* and *d*, visual inspection suggests that the simulation that estimates the locations provides a better fit than the simulation with free weights. To prevent unnecessary flexibility, we will, therefore, estimate a factor representing the pull exerted on the component distributions, yet not estimate the relative weights of the components.

## 3.3. Fitting the mild cognitive impaired sample

After confirming that the MLN model matches the data of the young-adult population and the more complex empirical priors described in Acerbi *et al*. [12], we now assess whether this model can provide additional insight in empirical phenomena and, vice versa, whether the parameters associated with the models relate in a sensible manner to external constructs. Hereto, we fitted the MLN model to the clinical data collected by Maaß *et al*. [22]. In this work, we demonstrated that participants diagnosed with MCI demonstrated a stronger central tendency effect than age-matched, healthy control (HC) participants (see for a detailed description of these populations [22]). This result might seem paradoxical as MCI status is partially defined by memory dysfunction, whereas a stronger central tendency effect suggests an emphasized influence of memory. By fitting the MLN model to

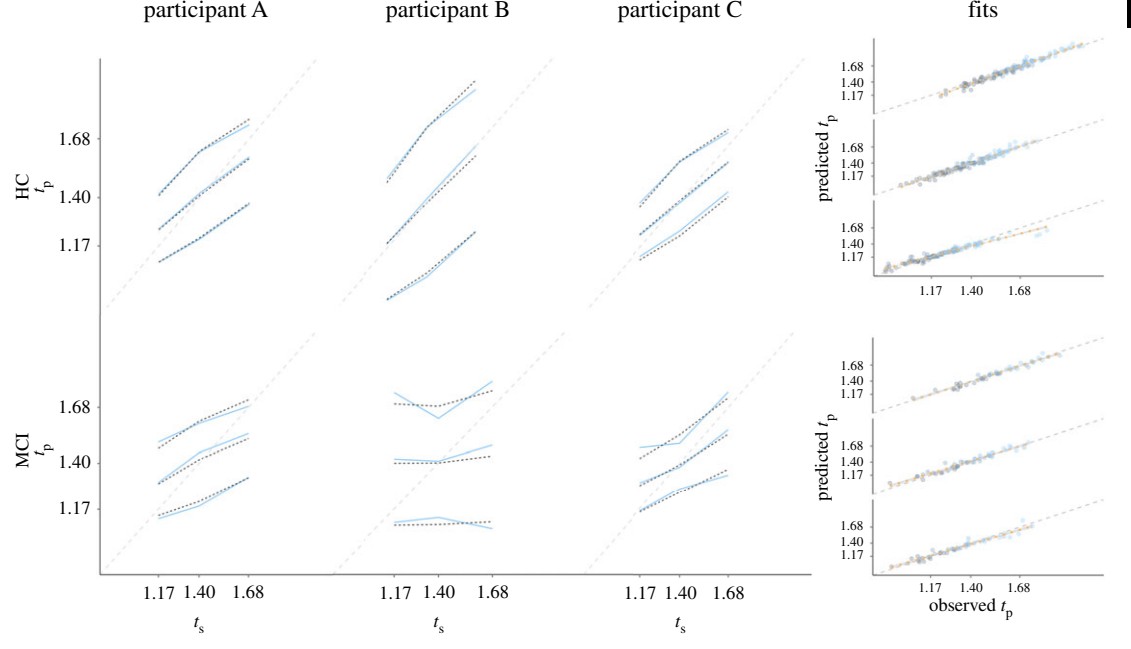

**Figure 6.** Fit of the MCI model to the empirical data from Maaß *et al.* [22]. The three left-most columns depict the 10, 50 and 90% quantile of the empirical data (blue) and MLN model predictions (black) for three example participants ($t_s$: presented interval; $t_p$: produced interval in seconds). The two rows represent the two participant groups (MCI and HC participants). The right-most panel depicts, for each of the three $t_s$ durations, the difference between predicted quantiles and the observed empirical quantiles for all HC ($n = 25$) and MCI ($n = 10$) participants, with colour scale representing the 10 (black), 30, 50, 70 and 90% (blue) quantiles.

this data, we hope to elucidate the locus of the increased central tendency effect by assessing whether a difference can be observed in the priors associated with both participant groups, or whether this effect is driven by noisier clock systems. Figure 6, mirroring the layout of figure 3, presents the fits of the MLN model to three MCI (top row) and three control (bottom row) participants. The right-most column again shows the model fit for all participants, again demonstrating that the deviation between predicted and observed responses is relatively small for all quantiles. Figure 7 plots the distributions of the four parameters of the MLN model for the MCI and the HC participants, with in the top-left corner the Bayes factor representing the evidence in favour of a difference between both distributions as a function of MCI status (assessed with lmBF, [50]). Importantly, while there is a difference (BF = 7.76) observed between both groups for $w_m$ (internal clock noise), there is no evidence for a difference for $w_p$ between both groups (but note that the BF = 0.58 in figure 7 does not meet the typical thresholds for considering results to be reliably interpretable, [54]). This finding is in line with earlier work that has demonstrated that clock noise increases as a function of cognitive decline [55–61]. However, and most notably, this contrasts our earlier report in which we argued that there was no difference between both groups in terms of clock noise when we determined clock noise using the 1 s task.

In the 1 s task, part of both datasets reported in this manuscript [21,22], participants are asked to produce a duration of 1 s repeatedly by means of a keypress. In our earlier work, we have argued that the measure derived from this task 'is related to the width of the likelihood distribution in the multi-duration reproduction task, which has been associated with the noise in the clock parts of the temporal system' [21, p. 8]. This predicts, in terms of the discussed Bayesian observer models, that the observed measure should correlate with $w_m$ as this measure indexes clock noise, but also with $w_p$ in the Uniform and Gaussian models, as $w_p$ reflects both clock and motor noise in these models. However, shown in the columns for the Uniform and Gaussian models of table 2, the only reliable correlation is found between $w_m$ and the 1 s task for the Uniform model, but no reliable results are observed for $w_p$ in either model. Instead, the distribution width parameter of the Gaussian model ($p_{sd}$) is negatively correlated to the 1 s variance, indicating that a higher variance during the 1 s task results in stronger pull effects due to a stronger prior influence. Obviously, this pattern results in the empirical observation of the stronger pull for participants with higher 1 s variance measures [21,22], but it is difficult to conceive a mechanism that can both explain an increased 1 s production variance while at the same time *only* predicting a more narrow prior and no reliable clock-noise effects.

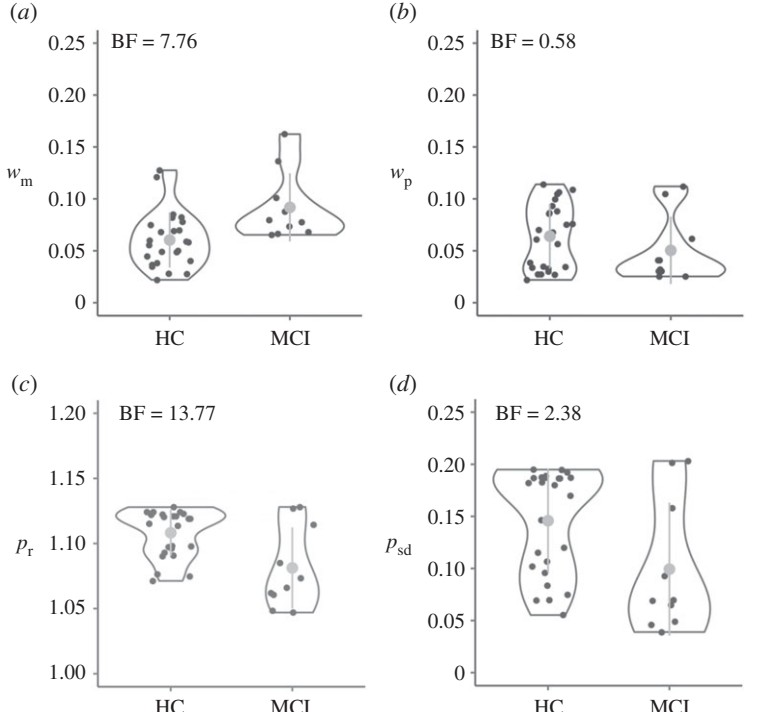

**Figure 7.** MLN model parameters for the HC and MCI groups from Maaß et al. [22]. Violin and whisker plot (representing mean and two standard deviations) represent model parameters estimated by means of grid search and Bayesian model averaging for $w_m$, $w_p$, $p_r$ and $p_{sd}$ by a participant group. Dots ( jittered for visualization) represent individual participants. The BF value in the top-left of each panel reflects the Bayes factor in favour of participant group predicting that panel's model parameter determined by the lmBF function of the BayesFactor package [50].

**Table 2.** Correlations between 1 s task measure to predict clock noise (see [21]) and the model parameters. The $w_p$/$\Delta w_p$ row lists $w_p$ for the Uniform and Gaussian models, and $\Delta w_p$ for the MLN model. The $r$ column lists Pearson's product moment correlation coefficient, the BF column lists the Bayes factor determined by Jeffreys' [51] test for linear correlation as implemented in the correlation BF function of the BayesFactor package [50].

| model | Maaß & Van Rijn [21] ($n = 57$, young healthy adults) | | | | | | | | Maaß et al. [22] ($n = 30$, MCI and HC) | |
| | Uniform | | Gaussian | | | MLN | | | MLN | |
| | $r$ | BF | $r$ | BF | $r$ | BF | $r$ | BF | $r$ | BF |
|---|---|---|---|---|---|---|---|---|---|---|
| $w_m$ | 0.38 | 16.247 | 0.21 | 0.907 | 0.32 | 5.046 | | | 0.19 | 0.622 |
| $w_p$/$\Delta w_p$ | 0.20 | 0.845 | 0.26 | 1.885 | 0.11 | 0.403 | | | 0.33 | 1.649 |
| $p_{sd}$ | | | −0.39 | 22.936 | −0.19 | 0.763 | | | 0.36 | 2.210 |
| $p_r$ | | | | | −0.15 | 0.546 | | | 0.24 | 0.837 |

Interestingly, the MLN model does show a correlation between 1 s variability and $w_m$, but not between 1 s variability and $\Delta w_p$. As the latter parameter represents motor noise, this pattern of results supports the notions forwarded in Maaß & Van Rijn [21] that the 1 s variability measure captures clock noise, and provides conceptual support for the internal validity of the MLN model. However, no reliable relations can be found for the MCI sample. This could be due to the smaller number of participants, or because this sample consists of multiple subsamples (i.e. healthy aged, non-diagnosed but memory-affected, and MCI-diagnosed participants, [22]). As analysing these subsamples did not provide qualitatively different results, we refrain from interpretation, but recommend reassessing this

relation when data of a larger sample of participants are available. Obviously, an alternative explanation is that the results of the 1 s task are influenced by motor noise, which might be stronger in the aged population than in the young-adult population. The results reported in table 2 hint in that direction, as where there is positive evidence for $w_m$ and negative evidence for $w_p$ relating to 1 s production noise in the young-adult population, the effects are reversed for the aged population. This argues that stronger effects of motor noise in aged populations could explain the disparity between the results in figure 7a (where we find $w_m$ differences between MCI and HC participants) and the earlier reported [22] absence of clock noise differences between these groups.

Apart from the difference between MCI and HC for the $w_m$ parameter, figure 7c also depicts a difference for the $p_r$ parameter which reflects the spread of the individual components contributing to the MLN prior. The smaller $p_r$ values estimated for the MCI participants indicate that their prior is more compact, resulting in a stronger memory-based central tendency effect. This is in line with the conclusions drawn on the statistical analysis of the behavioural data reported in Maaß et al. [22], and thus provides additional support for the hypothesis that in this memory-affected population the internal representation of earlier experiences is weighted more strongly than in HC populations. Whereas Bayesian observer models are typically considered to represent optimally as the integration with the prior compensates for a lack of accuracy, in this clinical population compensation might be needed to counter the decay of the memory trace of the current interval.

# 4. General discussion

We demonstrated and argued that Bayesian observer models that allow for interindividual variability in the shape of the prior outperform a Bayesian observer model that assumes a Uniform prior, but that only the MLN model adheres to theoretical constraints regarding perception and production noise. In addition, this more realistic MLN model provides new insights regarding the differences between MCI patients and HC in a context task. Whereas, in our original work, we attributed the increased context effects uniquely to a more narrow prior for MCI patients, the computational cognitive model demonstrated that the combination of a narrower prior *and* a noisier internal clock drive the observed patterns. The components that make up the prior are more strongly pulled inwards, rather than represented by a narrower distribution (as would be the case in a Gaussian model). These results, thus, refine the conclusions that were drawn on the basis of a statistical analysis of the behavioural data ([22], see also [62]) and exemplifies the observation by Paraskevoudi et al. [63] that insight in deficient timing processes hinges upon a formalized approach that allows for dissecting whether memory or clock-noise processes drive deviations in timing performance.

Even though we presented the Gaussian and MLN models as two distinct models, the former could be seen as a special case of the latter. That is, apart from the distributional differences between a Gaussian and lognormal, the Gaussian model can be approximated by an MLN model with $p_r = 1$, as this would assume completely overlapping mixture prior distributions. However, the model simulations estimated $p_r$ to be 1.05 or higher, indicating that even in the participants with the strongest prior-based central tendency effect, the components were not fully overlapping (i.e. a $p_r = 1.05$ results in distribution means of 1.33, 1.4 and 1.47). Consequently, the estimated distributions are more leptokurtic than what would be obtained with a normal distribution.

In addition, the Gaussian distribution has been shown to fail to predict empirical patterns observed in studies that use a dense representation of intervals. In a number of such studies, it has been demonstrated that the central tendency effect may not be a linear deviation from the target duration, but rather follows a more sigmoid pattern, where more extreme data points gravitate towards the mean more than less extreme data points [15,64]. Narain et al. [64] demonstrated that the sigmoid pattern is conditional on a uniform prior, as a Gaussian prior does not yield sufficient curvature. Our work extends the observations forwarded by Narain et al., as it supports the observation that a prior distribution with a sudden transition from negligible prior density to a higher prior density yields a sigmoid pattern in central tendency. In the extreme case, when many underlying components are assumed to contribute to the prior, the resulting MLN prior distribution mimics a uniform distribution (figure 8a). The degree of mimicry to the uniform prior depends on the width of each component, with narrower components resulting in a distribution that more closely resembles the step-shaped boundaries of the uniform distribution closer. When wider components are assumed, the resulting prior more resembles a Gaussian distribution, yielding a weakening of the sigmoidal pattern (figure 8b). However, a high density of intervals is not required to observe the sigmoidal pattern. When, following the

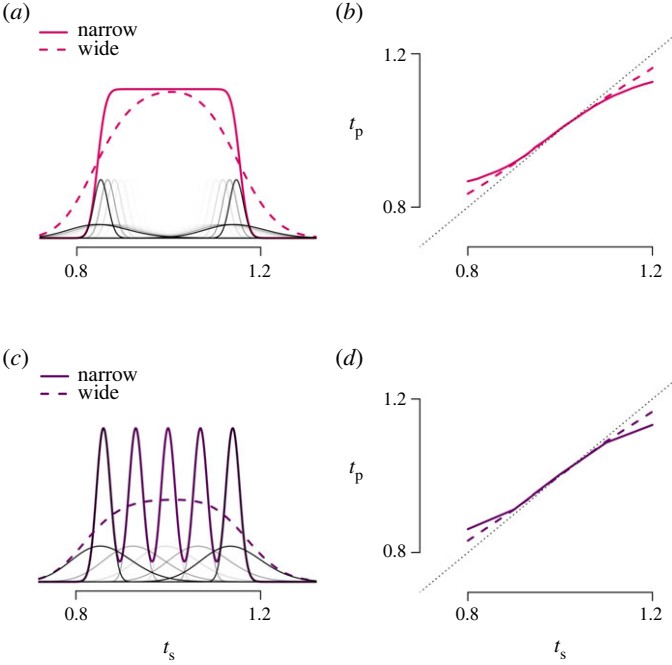

**Figure 8.** Simulations with the MLN model show that the model predicts complex behavioural patterns. Simulations plotted in the top row are based on an MLN prior consisting of 21 equidistant mixture components. With a sufficiently low standard deviation (narrow components drawn in black/grey lines), the resulting mixture resembles a uniform distribution (solid line). A larger standard deviation (wide components) results in a more unimodal distribution (dashed line). The right panel depicts that an increased width of the components results in increased nonlinearity in the central tendency effect. The simulations in the bottom row are based on an MLN prior consisting of five mixture components (similar to [15]). If the components have a sufficiently low standard deviation, the resulting mixture has a density that is negligible outside the stimulus range (solid purple line). A larger standard deviation (wide components) results in a more unimodal distribution (dashed line). Again, the different shapes of the MLN prior predict differences in the central tendency effect.

experimental paradigm of Sohn *et al*. [15], a smaller number of underlying components (figure 8*c*) is assumed, the MLN model still results in sigmoidal response patterns (figure 8*d*), the strength of which is again a function of the width of the mixture components. Thus, the MLN model reveals that it is not the uniform-like prior distribution *per se* that yields the sigmoid pattern in central tendency, but a sudden transition from negligible to increased prior densities. Moreover, it suggests that the strength of the sigmoid pattern appears as a result of individual differences in timing behaviour: individuals with more precise timing capabilities may develop prior distributions that have narrower mixture components, required for a sudden transition of prior density to demonstrate strong curvature effects.

One challenge when comparing different computational cognitive models, or variants of a single computational cognitive model, is to account for the degrees of freedom provided by additional parameters or assumed processing steps [65,66]. The same holds for the three models presented in this study, as it is not straightforward to provide an objective assessment of the exact number of degrees of freedom for each model. One could, for example, consider the parameters determining the width of a uniform prior as free-but-not-manipulated parameters in the Uniform model (cf. [67,68]): one could imagine a uniform prior to either extend beyond the maximum range of the presented durations if one assumed that a highly noisy internal clock blends out the perceived durations, or assume a more narrow uniform prior if one assumes that the estimated (or reproduced) durations would provide the basis for the prior. One could argue that this latter assumption is more in line with

Another test of the necessity of the added complexity of the MLN model is to test more extreme distributions, either in terms of the temporal distance between the presented standards, or by means of non-equal proportions of presented durations (figure 5). In both cases, the MLN model's fit to the data should improve relative to the Gaussian model, as this later model cannot capture the multi-dimensionality of the prior that would be induced by wider spaced durations, or unequal proportions of duration.

Bayesian reasoning as it fits well with the theoretical assumption that the model's posterior on a current trial contributes to the prior on future trials [10]. As the posterior will demonstrate the central tendency effect, one could hypothesize that the prior should also be narrower. Simply iterating over the number of estimated parameters is also problematic when considering $w_m$ and $w_p$. All three models assume two parameters for the noise associated with the perception and production stages, but only the MLN model constrains the latter to be larger than the former, reducing the model's flexibility.

A principled way to handle these issues associated with model flexibility is to estimate model parameters through Markov chain Monte Carlo (MCMC, e.g. [69]). This method would allow the quantification of uncertainty measures associated with the parameter estimates, because MCMC provides a (posterior) distribution of each parameter instead of a point estimate. Uncertainty in the parameter estimates is reflected in the predictive performance of each model, allowing for a fairer assessment of the trade-off between model flexibility and goodness-of-fit by way of computing Bayes factors comparing pairs of models. Such an approach would additionally allow for the comparison of model parameter distributions between groups in a straightforward manner (see, e.g. [70]).

An aspect that we deem more important than the statistical comparison of models of potentially different complexity is the potential interpretability of the cognitive models under consideration (that we refer to as construct validity). Even though the mathematical simplicity of the Uniform and Gaussian models could be taken as arguments in favour of their adoption, and both models fare well in cases of well-designed, balanced and simple paradigms, neither model allows for capturing more complex behavioural patterns that can readily be observed in empirical studies [12]. Moreover, especially in the case of the Uniform model as implemented here, central tendency effects are defined as a function of clock noise, as only by increasing the width of the likelihood can this model explain stronger central tendency effects. This rules out the possibility that memory-based explanations drive observed central tendency effects, which we argue is an important explanatory variable when considering the behavioural patterns observed in clinical populations. Additionally, a fixed prior also means that in the case of a strong central tendency effect but limited reproduction variance, the Uniform model has to assume a highly variable clock for the perception phase to account for the strong central tendency effect, but a very precise clock for the reproduction phase to capture the lack of variance. This dissociation between perception and reproduction clock noise is, to our knowledge, not supported by any empirical evidence. In sum, the models presented in this manuscript provide a hierarchy of conceptually plausible models and demonstrate that the different components of the MLN model provide the best tools to understand the clock and memory mechanisms involved in interval timing. Furthermore, as the basic tenet of this model is to provide a principled explanation of the underlying mechanisms of the central tendency effect, this work can be applied to any psychophysiological domain in which prior presentations influence the current percept (cf. [71]).

An important question that remains unanswered relates to how the cognitive system distinguishes between presented durations that are sampled from a continuous range and presented durations that are sampled from a fixed set of distinct durations. The latter case represents a typical interval timing experiment, including the experiments by Jazayeri & Shadlen [5], but the former case may be closer to a more natural updating of prior information about time intervals. From a pure Bayesian observer perspective, this is difficult to explain. However, in earlier work, Taatgen & Van Rijn [25] have demonstrated that an instance-based memory model can explain similar phenomena as the models presented in this manuscript. In this model, earlier experiences are stored in separate traces and a blending mechanism integrates these traces to result in the observed central tendency effects. When sufficient traces are presented of (very) similar durations, the blending process could cluster the traces together, resulting in a process-model analogue of the multiple distributions underlying the MLN model.

## 5. Conclusion

To conclude, a Bayesian observer model that assumes a prior that consists of a mixture of lognormal distributions outperforms a Bayesian observer model based on a uniform prior, and can fit *and* explain a number of empirical phenomena not captured by either a model based on a uniform or Gaussian prior. The MLN model does not just explain how Bayesian integration could take place, but also provides a theoretically sensible foundation for the shape of the prior. By fitting the prior to the observed behaviour, the MLN model provides a mechanistic account of how memory influences Bayesian integration. In addition, this model allows for a functional separation of clock noise, which is mostly driving the noise during the perception phase, and motor noise, which is an additional noise

component during the reproduction phase. With respect to the distribution of the prior, specific parametrizations of the MLN prior resemble the uniform and Gaussian priors that have been proposed earlier, indicating that the MLN model can capture the phenomena earlier ascribed to these more specific models while adhering to more stringent construct validity criteria. Thus, even though the MLN model is more complex than the existing models, theoretical and empirical considerations justify this model over the simpler models. Additionally, we demonstrated that this MLN model allows for a more precise interpretation of the behavioural results in a clinical population, paving the way for the utilization of computational cognitive models (cf. [72]) to assess the relative contribution of memory and clock components in declining performance in clinical populations. Lastly, this work also exemplifies the view that fit and complexity might come second to the interpretability of inferences as a criterion for choosing between models.

Data accessibility. The datasets analysed for this study can be found in the Open Science Framework (https://osf.io/kqjxf/).

Authors' contributions. S.C.M., L.v.M. and H.v.R. contributed to the development of ideas and the theoretical underpinnings, all authors contributed to the modelling of the data, and to the preparation of the manuscript.

Competing interests. We have no competing interests.

Funding. S.C.M., J.d.J. and H.v.R. were supported by the research program 'Interval Timing in the Real World: A functional, computational and neuroscience approach' with project no. 453-16-005, awarded to H.v.R., which is financed by The Netherlands Organization for Scientific Research (NWO).

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
