## [Peer Review File · Royal Society Open Science]

Review History

RSOS-201844.R0 (Original submission)

Review form: Reviewer 1 (Wolf Vanpaemel)

Is the manuscript scientifically sound in its present form?

Yes

Are the interpretations and conclusions justified by the results?

Yes

Is the language acceptable?

Yes

Do you have any ethical concerns with this paper?

No

Have you any concerns about statistical analyses in this paper?

No

Recommendation?

Accept with minor revision (please list in comments)

Comments to the Author(s)

The paper compares two existing models with different prior choices and proposes a third model with a more principled prior. The first two priors seem to be convenience priors, whereas the third has some theoretical appeal. I have argued before that priors should reflect theory (see e.g., Vanpaemel and Lee, 2012 https://ppw.kuleuven.be/okp/_pdf/Vanpaemel2012UPTFT.pdf) so I am very sympathetic to the push for replacing convenience priors with theoretical ones. Second, the authors propose a theoretically motivated constraint on two parameters, which is again in line with a general push for increasing the informativeness (and potentially the falsifiability) of the model.

I am an outsider to the field of perception, so I am afraid I can not offer a lot of helpful suggestions. (I accepted the review assignment hastily, thinking that it was on a different topic, where Bayes is used in a statistical sense rather than in the Bayes-in-the-head approach).

I did wonder whether it is interesting to compare the following models explicitly, to tease apart the influence of the two proposals (prior and constraint): Uniform (unconstrained) Gaussian (unconstrained) MLN (unconstrained) Uniform (constrained) Gaussian (constrained) MLN (constrained) instead of the current three models under (explicit) consideration. Some of these comparisons (but not all) are discussed verbally, but I think a more systematic and explicit approach might be interesting.

The theoretical considerations, model fits, and selective influence test speak to the validity of the new MLN model (although the small difference in RMSE between Gauss and MLN makes it a bit less compelling, especially given the nested nature between the two). Especially the surprising interpretation of the MCI sample speaks to the model. The necessity of the MLN model could further be increased by showing what interpretation the Gaussian model would provide for this data set.

For future work, I was wondering whether divergent predictions could be derived between the MLN and the Gaussian model. Making (and confirming) these predictions could further drive home the point that the additional complexity introduced by MLN is needed.

The ms is a bit repetitive at times (e.g., about the model implementation of how (not) to constrain an and wp). I think the readability can be improved by avoiding this needless repetition.

Huys et al., 2016 was missing in the references.

signed
wolf vanpaemel

Review form: Reviewer 2

Is the manuscript scientifically sound in its present form?

Yes

Are the interpretations and conclusions justified by the results?

Yes

Is the language acceptable?

Yes

Do you have any ethical concerns with this paper?

No

Have you any concerns about statistical analyses in this paper?

No

Recommendation?

Major revision is needed (please make suggestions in comments)

Comments to the Author(s)

I am not an expert in perceptual decision making, and certainly do not know the interval timing literature. The basic idea of extending the flexibility of priors in an ideal observer modeling framework to include mixture components makes sense. Of course, the key is to justify this flexibility in terms of the basic modeling goals of description, explanation, and prediction. The paper focuses on explanation, and makes a case I found convincing enough, but I would defer to experts on this.

If that basic hurdle -- there is evidence for the utility of the MLN extension is met -- then I have a set of comments (major, minor, very minor) on Bayesian modeling and inference issues that may help improve the paper in revision. They are list below. I hope they are helpful.

MAJOR POINTS

I was underwhelmed by the grid-search-based point-estimate model fitting. Actually, the grid-search is fine -- I don't care what computational methods are used -- it is the optimization rather than inference, and point estimates rather than posteriors, which represent the conceptual parts of fitting that I was underwhelmed by. It is ironic in a paper that elegantly uses Bayesian machinery to understand how the visual system solves inference problems (analysis of sparse and noisy data in structured environments with an inductive bias provided by prior experience) that the same machinery is not used to analyze the models. It's the same type of problem. As scientists, the authors now have sparse and noisy behavioral data by which to evaluate structured perceptual decision-making models about which they have prior knowledge. Why move away from coherent, complete, and principled Bayesian inference when you need to do scientific inference? For example, in the Discussion the claim is made that "as it is not straightforward to provide an objective assessment of the exact number of degrees of freedom for each model" my reaction was to wonder why prior predictive performance (underlying Bayes factors) did not provide such an assessment. Relatedly, I think conceiving of complexity in terms of "number of parameters" or "degrees of freedom" is unhelpful here (and in general). The complexity of a model relates to how many predictions it can make, and especially whether it makes falsifiable predictions. Thinking of complexity in terms of number of parameters and degrees of freedom is often unhelpful, and I think that is the case here.

Section 3(b): The argument that fit and complexity can be ignored to focus on the interpretability of inferences as a criterion for choosing between models is interesting. I agree with the authors that interpretability (e.g., the selective influence of experimental manipulations on model parameters) is very important, and not widely enough appreciated or used in psychological modeling. Too often the goal is just to optimize goodness-of-fit against complexity, and then only crudely. But, I think goodness-of-fit still matters, because unless a model is descriptively adequate, all bets are off. A model that had highly interpretable parameter inferences but couldn't re-describe the data on which they were based in not a good model, I would claim. And I think complexity still matters too, because it is a gate-keeper to good prediction. Overall, I often think of models as having three goals -- description, explanation, and prediction. I would say they go in that order. You need description. You want explanation, which is the focus here. Ideally, you'd

have prediction too. I certainly buy the argument here that a model that describes and predicts as well as a rival, and offers a more intuitive explanation, is superior.

Section 3(c): I was confused by this short section. Are AIC weights being used to approximate BFs (or posterior odds?) as part of some model averaging. How is the Bayes Factor package being used if AIC weights are already being used? The use of the BFs in the later results sections (and the conceptual framework for their interpretation provided in this section) both seem solid, but I just couldn't figure out how they had been estimated. Was the Bayes Factor package just used to compare parameter estimates across subject groups, as in JASP?

MINOR POINTS

- the abstract mentioning interval timing tasks as an example, but it wasn't clear to me by the end of the abstract whether the paper's applications of the MLN were also going to be to this task, or some other perceptual task. I imagine many readers skimming abstracts would like this key bit of information as to what tasks are considered in the paper.

- Figure 1 is very hard to read in the pdf I was sent. It looks to be bitmapped rather than raster, and downsampled. The actual figure content and design is nice, but it took a lot of zooming and squinting to read numbers, spot very faint distributions, etc.

- I'm not familiar with earlier work in this specific area. (I do know closely-related Bayesian observer models for size estimation in the category structure literature -- Huttenlocher, Hemmer & Steyers, ...). I was surprised by the original choice of uniform (on both psychological plausibility and analytic convenience grounds). The use of Normals was expected. The motivation in a timing domain for non-zero right-skewed distributions makes sense, and the ones listed "the Wald or inverse Gaussian, skewed-Normal, or log-Normal distribution" have all been taken seriously in, for example, the response time modeling literature, along with others (e.g., the Weibull). Is there any theoretical or empirical justification for the current focus on the log-normal?

- Does Figure 2 shows means over all subjects with the points, and standard errors of those means with error bars? If that was made explicit, I missed it. If it is standard errors, then I would be interested to know more about individual differences. There are many ways the standard error of means for MCI could be larger (a few subjects vary different, variability across all subjects, just lower numbers), and it would be interesting to know which (or which combination) is the case.

- "We assume that the number of mixture components equals the number of empirically presented durations." This is doable, and sensible enough, in a controlled experiment where the stimulus environment is known. How does the number of mixture components in the MLN scale beyond controlled settings?

- I would suggest always expressing BFs as ratios > 1 , and indicate which hypothesis they favor (e.g., $BF = 0.05$ in favor of difference re-expressed as $BF = 20$ in favor of sameness).

- Figure 3's caption says "The three leftmost columns depict the 10%, 50%, and 90% quantile (in blue) and model predictions (in black) ..." But those are not model predictions, right? (It would be remarkable if they were). They are based on the model already having the data, and use the parameter inferences that come from that knowledge, right? If so, they are definitely not predictions, and it's very misleading to call them that, I would call them descriptions. You could call them "fits". (I hate that word, but that's me).

- The section on empirically (and non-parametrically) determined priors was very interesting.

- How was MCI determined for this sample? The results in Figure 7 almost suggest bi-modality. I'm not an expert, but I believe things like this can happen in some functional assessments of MCI, which focus on living skills and so on. One way people can end up classified as impaired is through memory deterioration (the focus here, and what presumably causes the worse task performance), but it is also possible to be classified as functionally impaired because of, for example, depression, which does not involve cognitive deficits that might be expected to influence a timing interval task.

VERY MINOR THINGS

- "First, we will fit the three models to the data of a sample of 68 healthy young adults (undergraduate students). We determined ..." changes tenses pretty jarringly.

- The ms sometimes splits infinitives (e.g., "to accurately reproduce", "to partially capture") which, for some readers (like me), will grate

Review form: Reviewer 3

Is the manuscript scientifically sound in its present form?

Yes

Are the interpretations and conclusions justified by the results?

Yes

Is the language acceptable?

Yes

Do you have any ethical concerns with this paper?

No

Have you any concerns about statistical analyses in this paper?

No

Recommendation?

Accept with minor revision (please list in comments)

Comments to the Author(s)

In the paper the authors compared three Bayesian inference models, which differ in the shape of the prior, for interval reproductions: The uniform prior proposed by Jazayeri and Shadlen (2010), Gaussian prior by Cicchini et al. (2012), and their mixture lognormal (MLN) prior. To compare the predictions, they fit three models to the published interval reproduction data from healthy participants as well as clinical populations of aged mild cognitive impairment (MCI) patients, and demonstrate that their model fits better in certain aspects. The new MLN model is certainly interesting and contributes the ongoing development of Bayesian inference in time perception. Overall I like this approach and their interpretation of the MCI data. I do however have some conceptual and more specific comments that I think would improve the manuscript.

1. The justification of using mixture lognormal distribution as the prior (Shape of the prior): The authors argued that response functions in timing studies are better described by skewed distribution (p. 4, second paragraph), so the skewed prior is an ideal candidate. I think the skewed response function is the final output, while the prior often refers to an internal statistical representation of the sampled intervals. Whether the distribution of the response function can be treated as the distribution for the prior needs a bit more justification. Perhaps the authors could look into one key signature of interval timing - scalar property, which links to Weber's law and may suggest a potential internal logarithmic encoding of time intervals. When a Gaussian prior is transformed from internal logarithmic scale to external linear scale, it becomes a lognormal distribution. Thus, whether using lognormal (a skewed shape) or normal distribution (as suggested by Cicchini et al., 2012) as the prior depends on at which scale you apply Bayesian integration. A recent paper Ren et al. (2021) used a model with an internal Gaussian prior at log-scale, which can also predict a lognormal distribution of the reproduction.

2. p4. the final paragraph. I do not fully understand the logic here. My understanding is that if you model the prior with an internal logarithmic scale, you might have a simpler Gaussian prior with the scalar property and the skewed central tendency (biased more with the long relative to the short duration).

3. Model comparisons based on tp only

pp. 8-9 Fitting the student sample: Here three models were compared based on the reproduction (tp), but the differences in terms of RMSEs are relative small (0.032, 0.028 and 0.027, respectively). I wonder if the prediction from MLN is better than others for accounting both tp and the standard deviations of tp (i.e., predicting the distribution of tp). Or, in other words, would the MLN predicts better for the observed tp and the scalar property? Same comment can be applied for the model comparison of MCI data.

Minors

4. p4. line 51. "... the locations of the other components reflect ..." is not clear to me. "what other components do you refer to?"

5. p4. line 55-56. "We will assume that the means of the distributions will be distributed as a geometric series, resembling the type of distribution of the presented stimulus durations". It is not clear to me. Were your durations sampled in geometric spacing or are there any literature supporting this assumption?

6. p.6. Analysis code of Maass and Van Rijn (2018) is available, but the data are missing in the OSF repository.

7. p 12. line 51. "there is no evidence for a difference for w_p between both groups" This needs statistical support.

Reference:

Ren, Y., Allenmark, F., Müller, H. J., & Shi, Z. (2021). Variation in the "coefficient of variation": Rethinking the violation of the scalar property in time-duration judgments. *Acta Psychologica*, 214, 103263.

Decision letter (RSOS-201844.R0)

Dear Ms Maaß

The Editors assigned to your paper RSOS-201844 "Conceptually Plausible Bayesian Inference in Interval Timing" have now received comments from reviewers and would like you to revise the paper in accordance with the reviewer comments and any comments from the Editors. Please note this decision does not guarantee eventual acceptance.

Please submit your revised manuscript and required files (see below) no later than 21 days from today's (ie 04-Mar-2021) date. Note: the ScholarOne system will 'lock' if submission of the revision is attempted 21 or more days after the deadline. If you do not think you will be able to meet this deadline please contact the editorial office immediately.

on behalf of Professor Zoltan Dienes (Associate Editor) and Essi Viding (Subject Editor)
openscience@royalsociety.org

Associate Editor Comments to Author (Professor Zoltan Dienes):
Comments to the Author:

The three reviewers were overall positive about your work, but had detailed comments, including about the process of model selection. I was confused about exactly how the process worked. At a general level, I think the goals of explanation, prediction and accounting for

complexity are not entirely independent; that is, an explanation is not really explanatory if it can closely fit the data because it could fit most any data. Bayes factors are an ideal way of dealing with this issue. But then why use AIC weightings? Why not just see how well the data are predicted given the full range of allowed parameter values (with whatever prior constraints you justify - as Reviewer 1 complimented you for)? What exactly did you enter into the Bayes factor package to get your Bayes factors? The process seemed clear enough to two of the reviewers not to comment, but I would urge you to take Reviewer 2's points seriously in terms of considering the most coherent method of model selection.

Reviewer comments to Author:

Reviewer: 1

Comments to the Author(s)

The paper compares two existing models with different prior choices and proposes a third model with a more principled prior. The first two priors seem to be convenience priors, whereas the third has some theoretical appeal. I have argued before that priors should reflect theory (see e.g., Vanpaemel and Lee, 2012 https://ppw.kuleuven.be/okp/_pdf/Vanpaemel2012UPTFT.pdf) so I am very sympathetic to the push for replacing convenience priors with theoretical ones. Second, the authors propose a theoretically motivated constraint on two parameters, which is again in line with a general push for increasing the informativeness (and potentially the falsifiability) of the model.

I am an outsider to the field of perception, so I am afraid I can not offer a lot of helpful suggestions. (I accepted the review assignment hastily, thinking that it was on a different topic, where Bayes is used in a statistical sense rather than in the Bayes-in-the-head approach).

I did wonder whether it is interesting to compare the following models explicitly, to tease apart the influence of the two proposals (prior and constraint): Uniform (unconstrained) Gaussian (unconstrained) MLN (unconstrained) Uniform (constrained) Gaussian (constrained) MLN (constrained) instead of the current three models under (explicit) consideration. Some of these comparisons (but not all) are discussed verbally, but I think a more systematic and explicit approach might be interesting.

The theoretical considerations, model fits, and selective influence test speak to the validity of the new MLN model (although the small difference in RMSE between Gauss and MLN makes it a bit less compelling, especially given the nested nature between the two). Especially the surprising interpretation of the MCI sample speaks to the model. The necessity of the MLN model could further be increased by showing what interpretation the Gaussian model would provide for this data set.

For future work, I was wondering whether divergent predictions could be derived between the MLN and the Gaussian model. Making (and confirming) these predictions could further drive home the point that the additional complexity introduced by MLN is needed.

The ms is a bit repetitive at times (e.g., about the model implementation of how (not) to constrain an and wp). I think the readability can be improved by avoiding this needless repetition.

Huys et al., 2016 was missing in the references.

signed

wolf vanpaemel

Reviewer: 2

Comments to the Author(s)

I am not an expert in perceptual decision making, and certainly do not know the interval timing literature. The basic idea of extending the flexibility of priors in an ideal observer modeling framework to include mixture components makes sense. Of course, the key is to justify this flexibility in terms of the basic modeling goals of description, explanation, and prediction. The paper focuses on explanation, and makes a case I found convincing enough, but I would defer to experts on this.

If that basic hurdle -- there is evidence for the utility of the MLN extension is met -- then I have a set of comments (major, minor, very minor) on Bayesian modeling and inference issues that may help improve the paper in revision. They are list below. I hope they are helpful.

MAJOR POINTS

I was underwhelmed by the grid-search-based point-estimate model fitting. Actually, the grid-search is fine -- I don't care what computational methods are used -- it is the optimization rather than inference, and point estimates rather than posteriors, which represent the conceptual parts of fitting that I was underwhelmed by. It is ironic in a paper that elegantly uses Bayesian machinery to understand how the visual system solves inference problems (analysis of sparse and noisy data in structured environments with an inductive bias provided by prior experience) that the same machinery is not used to analyze the models. It's the same type of problem. As scientists, the authors now have sparse and noisy behavioral data by which to evaluate structured perceptual decision-making models about which they have prior knowledge. Why move away from coherent, complete, and principled Bayesian inference when you need to do scientific inference? For example, in the Discussion the claim is made that "as it is not straightforward to provide an objective assessment of the exact number of degrees of freedom for each model" my reaction was to wonder why prior predictive performance (underlying Bayes factors) did not provide such an assessment. Relatedly, I think conceiving of complexity in terms of "number of parameters" or "degrees of freedom" is unhelpful here (and in general). The complexity of a model relates to how many predictions it can make, and especially whether it makes falsifiable predictions. Thinking of complexity in terms of number of parameters and degrees of freedom is often unhelpful, and I think that is the case here.

Section 3(b): The argument that fit and complexity can be ignored to focus on the interpretability of inferences as a criterion for choosing between models is interesting. I agree with the authors that interpretability (e.g., the selective influence of experimental manipulations on model parameters) is very important, and not widely enough appreciated or used in psychological modeling. Too often the goal is just to optimize goodness-of-fit against complexity, and then only crudely. But, I think goodness-of-fit still matters, because unless a model is descriptively adequate, all bets are off. A model that had highly interpretable parameter inferences but couldn't re-describe the data on which they were based in not a good model, I would claim. And I think complexity still matters too, because it is a gate-keeper to good prediction. Overall, I often think of models as having three goals -- description, explanation, and prediction. I would say they go in that order. You need description. You want explanation, which is the focus here. Ideally, you'd have prediction too. I certainly buy the argument here that a model that describes and predicts as well as a rival, and offers a more intuitive explanation, is superior.

Section 3(c): I was confused by this short section. Are AIC weights being used to approximate BFs (or posterior odds?) as part of some model averaging. How is the Bayes Factor package being used if AIC weights are already being used? The use of the BFs in the later results sections (and the conceptual framework for their interpretation provided in this section) both seem solid, but I

just couldn't figure out how they had been estimated. Was the Bayes Factor package just used to compare parameter estimates across subject groups, as in JASP?

MINOR POINTS

- the abstract mentioning interval timing tasks as an example, but it wasn't clear to me by the end of the abstract whether the paper's applications of the MLN were also going to be to this task, or some other perceptual task. I imagine many readers skimming abstracts would like this key bit of information as to what tasks are considered in the paper.

- Figure 1 is very hard to read in the pdf I was sent. It looks to be bitmapped rather than raster, and downsampled. The actual figure content and design is nice, but it took a lot of zooming and squinting to read numbers, spot very faint distributions, etc.

- I'm not familiar with earlier work in this specific area. (I do know closely-related Bayesian observer models for size estimation in the category structure literature -- Huttenlocher, Hemmer & Steyers, ...). I was surprised by the original choice of uniform (on both psychological plausibility and analytic convenience grounds). The use of Normals was expected. The motivation in a timing domain for non-zero right-skewed distributions makes sense, and the ones listed "the Wald or inverse Gaussian, skewed-Normal, or log-Normal distribution" have all been taken seriously in, for example, the response time modeling literature, along with others (e.g., the Weibull). Is there any theoretical or empirical justification for the current focus on the log-normal?

- Does Figure 2 shows means over all subjects with the points, and standard errors of those means with error bars? If that was made explicit, I missed it. If it is standard errors, then I would be interested to know more about individual differences. There are many ways the standard error of means for MCI could be larger (a few subjects vary different, variability across all subjects, just lower numbers), and it would be interesting to know which (or which combination) is the case.

- "We assume that the number of mixture components equals the number of empirically presented durations." This is doable, and sensible enough, in a controlled experiment where the stimulus environment is known. How does the number of mixture components in the MLN scale beyond controlled settings?

- I would suggest always expressing BFs as ratios > 1 , and indicate which hypothesis they favor (e.g., $BF = 0.05$ in favor of difference re-expressed as $BF = 20$ in favor of sameness).

- Figure 3's caption says "The three leftmost columns depict the 10%, 50%, and 90% quantile (in blue) and model predictions (in black) ..." But those are not model predictions, right? (It would be remarkable if they were). They are based on the model already having the data, and use the parameter inferences that come from that knowledge, right? If so, they are definitely not predictions, and it's very misleading to call them that, I would call them descriptions. You could call them "fits". (I hate that word, but that's me).

- The section on empirically (and non-parametrically) determined priors was very interesting.

- How was MCI determined for this sample? The results in Figure 7 almost suggest bi-modality. I'm not an expert, but I believe things like this can happen in some functional assessments of MCI, which focus on living skills and so on. One way people can end up classified as impaired is through memory deterioration (the focus here, and what presumably causes the worse task performance), but it is also possible to be classified as functionally impaired because of, for

example, depression, which does not involve cognitive deficits that might be expected the influence a timing interval task.

VERY MINOR THINGS

- "First, we will fit the three models to the data of a sample of 68 healthy young adults (undergraduate students). We determined ..." changes tenses pretty jarringly.
- The ms sometimes splits infinitives (e.g., "to accurately reproduce", "to partially capture") which, for some readers (like me), will grate

Reviewer: 3

Comments to the Author(s)

In the paper the authors compared three Bayesian inference models, which differ in the shape of the prior, for interval reproductions: The uniform prior proposed by Jazayeri and Shadlen (2010), Gaussian prior by Cicchini et al. (2012), and their mixture lognormal (MLN) prior. To compare the predictions, they fit three models to the published interval reproduction data from healthy participants as well as clinical populations of aged mild cognitive impairment (MCI) patients, and demonstrate that their model fits better in certain aspects. The new MLN model is certainly interesting and contributes the ongoing development of Bayesian inference in time perception. Overall I like this approach and their interpretation of the MCI data. I do however have some conceptual and more specific comments that I think would improve the manuscript.

1. The justification of using mixture lognormal distribution as the prior (Shape of the prior): The authors argued that response functions in timing studies are better described by skewed distribution (p. 4, second paragraph), so the skewed prior is an ideal candidate. I think the skewed response function is the final output, while the prior often refers to an internal statistical representation of the sampled intervals. Whether the distribution of the response function can be treated as the distribution for the prior needs a bit more justification. Perhaps the authors could look into one key signature of interval timing - scalar property, which links to Weber's law and may suggest a potential internal logarithmic encoding of time intervals. When a Gaussian prior is transformed from internal logarithmic scale to external linear scale, it becomes a lognormal distribution. Thus, whether using lognormal (a skewed shape) or normal distribution (as suggested by Cicchini et al., 2012) as the prior depends on at which scale you apply Bayesian integration. A recent paper Ren et al. (2021) used a model with an internal Gaussian prior at log-scale, which can also predict a lognormal distribution of the reproduction.

2. p4. the final paragraph. I do not fully understand the logic here. My understanding is that if you model the prior with an internal logarithmic scale, you might have a simpler Gaussian prior with the scalar property and the skewed central tendency (biased more with the long relative to the short duration).

3. Model comparisons based on tp only

pp. 8-9 Fitting the student sample: Here three models were compared based on the reproduction (tp), but the differences in terms of RMSEs are relative small (0.032, 0.028 and 0.027, respectively). I wonder if the prediction from MLN is better than others for accounting both tp and the standard deviations of tp (i.e., predicting the distribution of tp). Or, in other words, would the MLN predicts better for the observed tp and the scalar property? Same comment can be applied for the model comparison of MCI data.

Minors

4. p4. line 51. "... the locations of the other components reflect ..." is not clear to me. "what other components do you refer to?"

5. p4. line 55-56. "We will assume that the means of the distributions will be distributed as a geometric series, resembling the type of distribution of the presented stimulus durations". It is not clear to me. Were your durations sampled in geometric spacing or are there any literature supporting this assumption?

6. p.6. Analysis code of Maass and Van Rijn (2018) is available, but the data are missing in the OSF repository.

7. p 12. line 51. "there is no evidence for a difference for w_p between both groups" This needs statistical support.

Reference:

Ren, Y., Allenmark, F., Müller, H. J., & Shi, Z. (2021). Variation in the "coefficient of variation": Rethinking the violation of the scalar property in time-duration judgments. *Acta Psychologica*, 214, 103263.

===PREPARING YOUR MANUSCRIPT===

===PREPARING YOUR REVISION IN SCHOLARONE===

Author's Response to Decision Letter for (RSOS-201844.R0)

See Appendix A.

Decision letter (RSOS-201844.R1)

Dear Dr Maaß,

It is a pleasure to accept your manuscript entitled "Conceptually Plausible Bayesian Inference in Interval Timing" in its current form for publication in Royal Society Open Science.

on behalf of Professor Zoltan Dienes (Associate Editor) and Essi Viding (Subject Editor)
openscience@royalsociety.org

Appendix A

Responses to the comments of the Associate Editor and Three Reviewers

Maaß, De Jong, Van Maanen, & Van Rijn

Associate Editor Comments:

Comments to the Author:

The three reviewers were overall positive about your work, but had detailed comments, including about the process of model selection. I was confused about exactly how the process worked. At a general level, I think the goals of explanation, prediction and accounting for complexity are not entirely independent; that is, an explanation is not really explanatory if it can closely fit the data because it could fit most any data. Bayes factors are an ideal way of dealing with this issue. But then why use AIC weightings? Why not just see how well the data are predicted given the full range of allowed parameter values (with whatever prior constraints you justify - as Reviewer 1 complimented you for)? What exactly did you enter into the Bayes factor package to get your Bayes factors? The process seemed clear enough to two of the reviewers not to comment, but I would urge you to take Reviewer 2's points seriously in terms of considering the most coherent method of model selection.

We would like to thank you for these comments, and we agree with your points about the difficulty of explaining phenomena with too powerful models. We hope that we have addressed these issues in the revised manuscript, and in our responses to the reviewers below. Specifically, with respect to the “full range of parameter values”, we would like to refer you to the simulations we present in our response to Reviewer 1, where we assess whether and how certain simulation choices we made influence the fit. We also updated the section on how we calculated the reported Bayes Factors to the Materials and Methods section.

Reviewer: 1

Comments to the Author(s)

The paper compares two existing models with different prior choices and proposes a third model with a more principled prior. The first two priors seem to be convenience priors, whereas the third has some theoretical appeal. I have argued before that priors should reflect theory (see e.g.,

Vanpaemel and Lee, 2012 https://ppw.kuleuven.be/okp/_pdf/Vanpaemel2012UPTFT.pdf) so I am very sympathetic to the push for replacing convenience priors with theoretical ones. Second, the authors propose a theoretically motivated constraint on two parameters, which is again in line with a general push for increasing the informativeness (and potentially the falsifiability) of the model.

We would like to thank the reviewer for these positive words, and are happy to see that he shares our interest in more principled modelling approaches. We have updated the text to provide a broader context to our work.

[..]

I did wonder whether it is interesting to compare the following models explicitly, to tease apart the influence of the two proposals (prior and constraint): Uniform (unconstrained) Gaussian (unconstrained) MLN (unconstrained) Uniform (constrained) Gaussian (constrained) MLN (constrained) instead of the current three models under (explicit) consideration. Some of these comparisons (but not all) are discussed verbally, but I think a more systematic and explicit approach might be interesting.

The reviewer is correct that our proposed model (MLN, with conceptual constraints on the noise parameters) differs from the earlier models in two aspects: (1) the shape of the prior distribution, and (2) the conceptual constraint). One approach would be to systematically present all variations of models, and iteratively test these models against simpler variants. However, one of the main goals of the current paper is to question the validity of the theoretical/conceptual assumptions in these earlier models, despite their good fit to the data. In that sense, we feel that including all possible model combinations in the paper would distract from that goal, since we would introduce three models that we on validity ground would not accept anyway (Uniform-constrained, Gaussian-constrained, MLN-unconstrained). For completeness, we did implement these models and computed how well they account for the data of 63 healthy young volunteers (Figure R1). For this purpose, we applied a cross-validation technique where we repeatedly (10 times) optimized model parameters based on RT quantiles from 80 % of the data points of every individual. The left panel shows the distribution of root-mean-square deviations of every quantile for every model prediction, based on RT quantiles from the 20% left-out data points. This way, the RMSD measure accounts for model complexity as well, as the parameters were determined on another subset of the data. From this left panel, it becomes clear that on average the Uniform models fit the data best (Uu and Uc for Uniform-constrained and unconstrained, respectively). This is probably due to overfitting in the case of the Gaussian and MLN models, as in the main manuscript we demonstrate that the MLN model (slightly) outperforms the other, theoretically less-preferred models.

From the right panel, it becomes clear that both aspects are needed. As can be seen in the *Mu* violin plot, also the MLN-unconstrained model suffers from the same drawback as the unconstrained Uniform (*Uu*) and Gaussian (*Gu*) prior models in that negative motor noise parameters are estimated. This demonstrates that just a different assumption about the prior will not suffice to warrant parameter estimates that are theoretically meaningful (ie above zero).

Figure R1. Model comparison of all possible models in terms of prior (U, G, M for Uniform, Gaussia, and MLN) and constraint on the W_p parameter (u, c for unconstrained/constrained). Left. Distribution of RMSD. Right: Parameter estimates for the difference between W_p and W_m (for the unconstrained models), and W_p (for the constrained models).

The theoretical considerations, model fits, and selective influence test speak to the validity of the new MLN model (although the small difference in RMSE between Gauss and MLN makes it a bit less compelling, especially given the nested nature between the two).

We are happy that the reviewer shares our enthusiasm for our approach. Even though the MLN and the Gaussian model could, in principle, resemble one another quite closely, the constraints on the parameters (for example in terms of the minimal distance between the component distributions of the mixture model) result in distinct, qualitative differences between the models.

Especially the surprising interpretation of the MCI sample speaks to the model. The necessity of the MLN model could further be increased by showing what interpretation the Gaussian model would provide for this data

We have indeed fitted all models to the MCI data as well, but decided against including said models in the paper as the overall image did not differ; all models fit relatively well, but only the MLN model provides a good fit while meeting a number of reasonable conceptual constraints. (See, for a similar argument, two responses above.)

For future work, I was wondering whether divergent predictions could be derived between the MLN and the Gaussian model. Making (and confirming) these predictions could further drive home the point that the additional complexity introduced by MLN is needed.

We have added a short section on these divergent predictions to the discussion.

The ms is a bit repetitive at times (e.g., about the model implementation of how (not) to constrain an and wp). I think the readability can be improved by avoiding this needless repetition.

We have carefully read over the manuscript, and removed or reformulated sections that we thought to be redundant. We hope to have improved the manuscript's readability.

Huys et al., 2016 was missing in the references.

Reference added.

signed

wolf vanpaemel

Reviewer: 2

Comments to the Author(s)

I am not an expert in perceptual decision making, and certainly do not know the interval timing literature. The basic idea of extending the flexibility of priors in an ideal observer modeling framework to include mixture components makes sense. Of course, the key is to justify this flexibility in terms of the basic modeling goals of description, explanation, and prediction. The paper focuses on explanation, and makes a case I found convincing enough, but I would defer to experts on this.

If that basic hurdle -- there is evidence for the utility of the MLN extension is met -- then I have a set of comments (major, minor, very minor) on Bayesian modeling and inference issues that may help improve the paper in revision. They are list below. I hope they are helpful.

We would like to thank Reviewer 2 for his helpful comments. In general, we agree with the general position that a fully integrated (principled) Bayesian model would be optimal. However, such a model is not straightforward to implement, and may not provide additional insight into the nature of interval timing. For this reason, we refrain from major changes in the cognitive

and statistical modeling presented here. Below, we have provided our views to the specific issues raised by the reviewer. In the General Discussion of the paper, we added a section to highlight the benefits of such an integrated model for inferential purposes.

MAJOR POINTS

I was underwhelmed by the grid-search-based point-estimate model fitting. Actually, the grid-search is fine -- I don't care what computational methods are used -- it is the optimization rather than inference, and point estimates rather than posteriors, which represent the conceptual parts of fitting that I was underwhelmed by. It is ironic in a paper that elegantly uses Bayesian machinery to understand how the visual system solves inference problems (analysis of sparse and noisy data in structured environments with an inductive bias provided by prior experience) that the same machinery is not used to analyze the models. It's the same type of problem.

We agree with the general notion forwarded by Reviewer 2. However, here we have presented a Bayesian Cognitive Model, not Bayesian Statistics-based evaluation of a complex cognitive task. Where in the latter case posteriors would definitely be more appropriate, we would argue that in this case point estimates do serve their purpose as it allows us to straightforwardly identify differences between groups. We have considered utilizing hierarchical Bayesian modelling in which we would assume two populations, with their own mean and variance for each parameter. Posterior estimates of the 'MCI' and 'non-MCI' populations are then shrunk, making differences clearer. However, this would complicate the presented work - without providing clear benefits for the scope of this paper.

As scientists, the authors now have sparse and noisy behavioral data by which to evaluate structured perceptual decision-making models about which they have prior knowledge. Why move away from coherent, complete, and principled Bayesian inference when you need to do scientific inference? For example, in the Discussion the claim is made that "as it is not straightforward to provide an objective assessment of the exact number of degrees of freedom for each model" my reaction was to wonder why prior predictive performance (underlying Bayes factors) did not provide such an assessment. Relatedly, I think conceiving of complexity in terms of "number of parameters" or "degrees of freedom" is unhelpful here (and in general). The complexity of a model relates to how many predictions it can make, and especially whether it makes falsifiable predictions. Thinking of complexity in terms of number of parameters and degrees of freedom is often unhelpful, and I think that is the case here.

We completely agree that expressing model complexity in terms of free parameters is, at best, suboptimal, and that the breadth of the potential explanatory power is a better assessment of a model's complexity. From a more pragmatic point of view, that probably won't matter much in this case, as we would assume that the MLN Model would still be evaluated as the most

“complex” model. This is why we have aimed for stressing the explanatory and conceptual power of this model of the alternative models, in our view, explanation supersedes complexity in this case. Additionally, to modify this work in line with the recommendation would require a major rewrite of the paper, which would make the paper lose its focus. In our opinion, this would better be suited to be covered in a new manuscript.

Section 3(b): The argument that fit and complexity can be ignored to focus on the interpretability of inferences as a criterion for choosing between models is interesting. I agree with the authors that interpretability (e.g., the selective influence of experimental manipulations on model parameters) is very important, and not widely enough appreciated or used in psychological modeling.

We thank the reviewer for stressing this, also in our view, very important message of the paper, and have emphasized this point in the discussion.

Too often the goal is just to optimize goodness-of-fit against complexity, and then only crudely. But, I think goodness-of-fit still matters, because unless a model is descriptively adequate, all bets are off. A model that had highly interpretable parameter inferences but couldn't re-describe the data on which they were based in not a good model, I would claim. And I think complexity still matters too, because it is a gate-keeper to good prediction. Overall, I often think of models as having three goals -- description, explanation, and prediction. I would say they go in that order. You need description. You want explanation, which is the focus here. Ideally, you'd have prediction too. I certainly buy the argument here that a model that describes and predicts as well as a rival, and offers a more intuitive explanation, is superior.

We, again, agree with this reviewer. In our view, the rationale presented in this manuscript is close to the rationale proposed in this comment by the reviewer.

Section 3(c): I was confused by this short section. Are AIC weights being used to approximate BFs (or posterior odds?) as part of some model averaging. How is the Bayes Factor package being used if AIC weights are already being used? The use of the BFs in the later results sections (and the conceptual framework for their interpretation provided in this section) both seem solid, but I just couldn't figure out how they had been estimated. Was the Bayes Factor package just used to compare parameter estimates across subject groups, as in JASP?

Indeed, we used the Bayes Factor package to compare parameter estimates across subjects groups. To clarify that this section does not relate to the model fitting and model comparison, we added a sentence to stress the inferential nature of these analyses.

MINOR POINTS

- the abstract mentioning interval timing tasks as an example, but it wasn't clear to me by the end of the abstract whether the paper's applications of the MLN were also going to be to this task, or some other perceptual task. I imagine many readers skimming abstracts would like this key bit of information as to what tasks are considered in the paper.

The abstract now indicates that the MLN model is indeed applied to interval timing data.

- Figure 1 is very hard to read in the pdf I was sent. It looks to be bitmapped rather than raster, and downsampled. The actual figure content and design is nice, but it took a lot of zooming and squinting to read numbers, spot very faint distributions, etc.

We double checked the resolution of the TIFF images that has to be uploaded to the submission portal, and hope that the current version is better readable.

- I'm not familiar with earlier work in this specific area. (I do know closely-related Bayesian observer models for size estimation in the category structure literature -- Huttenlocher, Hemmer & Steyers, ...). I was surprised by the original choice of uniform (on both psychological plausibility and analytic convenience grounds). The use of Normals was expected. The motivation in a timing domain for non-zero right-skewed distributions makes sense, and the ones listed "the Wald or inverse Gaussian, skewed-Normal, or log-Normal distribution" have all been taken seriously in, for example, the response time modeling literature, along with others (e.g., the Weibull). Is there any theoretical or empirical justification for the current focus on the log-normal?

The log-normal distribution has been proposed as a candidate for interval timing in a number of manuscripts that appeared in the interval timing field (as cited in the manuscript), but also in more mathematical focused literature. We now cite this latter literature in the manuscript (e.g., Aquino, et al., 2018) and provide a more extensive discussion of the role of log-transformation in the timing literature.

- Does Figure 2 shows means over all subjects with the points, and standard errors of those means with error bars? If that was made explicit, I missed it. If it is standard errors, then I would be interested to know more about individual differences. There are many ways the standard error of means for MCI could be larger (a few subjects vary different, variability across all subjects, just lower numbers), and it would be interesting to know which (or which combination) is the case.

We have updated the caption to indicate that the error bars are standard errors of the mean with the within-participants Cousineau-Morey correction applied. The difference in the standard errors is mainly due to the relatively small number of participants in the MCI group. It turned out that some MCI participants were not capable of following the task instructions, and for some other MCI patients, more than 25% of their responses were faster than 500 or slower than 2500ms. After excluding those participants, the remaining participants' data was - for example in terms of the proportion of excluded data points (~ 1.2%) - not qualitatively different from the healthy aged participants. More details about the selection criteria are reported in Maaß et al. (2019).

- "We assume that the number of mixture components equals the number of empirically presented durations." This is doable, and sensible enough, in a controlled experiment where the stimulus environment is known. How does the number of mixture components in the MLN scale beyond controlled settings?

This is an interesting question that is still currently open: How does the cognitive system distinguish between presented durations that are sampled from a continuous range, versus from a number of distinct durations? From a purely Bayesian Modeling perspective, this is difficult to explain. However, in earlier work, Taatgen and Van Rijn (2011) have demonstrated that an instance-based memory model can explain similar phenomena as the models presented in this manuscript. In this model, earlier experiences are stored in separate traces and a blending mechanism integrates these traces to result in the observed central tendency effects. When sufficient traces are presented of (very) similar durations, the blending process could cluster the traces together, resulting in a process-model analogue of the multiple distributions underlying the MLN model. We have added a paragraph that addresses this point.

- I would suggest always expressing BFs as ratios > 1 , and indicate which hypothesis they favor (e.g., $BF = 0.05$ in favor of difference re-expressed as $BF = 20$ in favor of sameness).

We have decided to leave the BF values as we originally presented them, as we are afraid that using different ratios in the figures would hinder interpretation. We have, however, where relevant added additional context to the text.

- Figure 3's caption says "The three leftmost columns depict the 10%, 50%, and 90% quantile (in blue) and model predictions (in black) ..." But those are not model predictions, right? (It would be remarkable if they were). They are based on the model already having the data, and use the parameter inferences that come from that knowledge, right? If so, they are definitely not predictions, and it's very misleading to call them that, I would call them descriptions. You could call them "fits". (I hate that word, but that's me).

Changed, even though we did opt for “fits”.

- The section on empirically (and non-parametrically) determined priors was very interesting.

Thank you for this compliment.

- How was MCI determined for this sample? The results in Figure 7 almost suggest bi-modality. I'm not an expert, but I believe things like this can happen in some functional assessments of MCI, which focus on living skills and so on. One way people can end up classified as impaired is through memory deterioration (the focus here, and what presumably causes the worse task performance), but it is also possible to be classified as functionally impaired because of, for example, depression, which does not involve cognitive deficits that might be expected to influence a timing interval task.

We more clearly refer the reader to the original manuscript, in which more detail is provided about this sample. In this case, participants did not have any significant comorbidities.

VERY MINOR THINGS

- "First, we will fit the three models to the data of a sample of 68 healthy young adults (undergraduate students). We determined ..." changes tenses pretty jarringly.

Changed.

- The ms sometimes splits infinitives (e.g., "to accurately reproduce", "to partially capture") which, for some readers (like me), will grate

Changed.

Reviewer: 3

Comments to the Author(s)

In the paper the authors compared three Bayesian inference models, which differ in the shape of the prior, for interval reproductions: The uniform prior proposed by Jazayeri and Shadlen (2010), Gaussian prior by Cicchini et al. (2012), and their mixture lognormal (MLN) prior. To compare the predictions, they fit three models to the published interval reproduction data from healthy participants as well as clinical populations of aged mild cognitive impairment (MCI) patients, and demonstrate that their model fits better in certain aspects. The new MLN model is certainly interesting and contributes the ongoing development of Bayesian inference in time perception. Overall I like this approach and their interpretation of the MCI data. I do however have some conceptual and more specific comments that I think would improve the manuscript.

We would like to thank the reviewer for the praise and the helpful comments, and will respond below to each of the comments raised.

1. The justification of using mixture lognormal distribution as the prior (Shape of the prior): The authors argued that response functions in timing studies are better described by skewed distribution (p. 4, second paragraph), so the skewed prior is an ideal candidate. I think the skewed response function is the final output, while the prior often refers to an internal statistical representation of the sampled intervals. Whether the distribution of the response function can be treated as the distribution for the prior needs a bit more justification.

We'd like to thank the reviewer for bringing up this critical issue. We realize that assuming the prior itself uses log-normally distributed representations (instead of those distributions being merely a behavioral output) needs a better justification than was provided in the previous version of the manuscript.

As the reviewer points out later, several Bayesian models of time perception have assumed a logarithmic encoding of time, in combination with Bayesian integration on that logarithmic scale. These models cleanly produce both the scalar property, log-normally distributed responses, and a stronger range effect for longer durations. Theoretically, the MLN model only departs in two important ways from these 'logarithmic' Bayesian models: (1) assumptions about the scalar property and (2) the scale at which Bayesian integration is applied: logarithmic or linear.

Anticipating our views on these matters, we feel that the theoretical differences are not always clear and relatively small, and that the empirical data we model in our study would not be able to arbitrate between the classic 'logarithmic' models and our MLN model. We will respond in more detail in the individual responses to your next comments.

We have included a brief discussion of these theoretical differences in the introduction on p4.

Perhaps the authors could look into one key signature of interval timing - scalar property, which links to Weber's law and may suggest a potential internal logarithmic encoding of time intervals. When a Gaussian prior is transformed from internal logarithmic scale to external linear scale, it becomes a lognormal distribution. Thus, whether using lognormal (a skewed shape) or normal distribution (as suggested by Cicchini et al., 2012) as the prior depends on at which scale you apply Bayesian integration. A recent paper Ren et al. (2021) used a model with an internal Gaussian prior at log-scale, which can also predict a lognormal distribution of the reproduction.

We agree with the reviewer that a logarithmic encoding of time naturally produces the scalar property (Ren et al., 2021). The MLN model, on the other hand, already assumes the scalar property in the likelihood function. But as the reviewer states, the scalar property only 'potentially suggests' a logarithmic encoding. There are theoretical proposals (Anobile et al., 2012) that first assume the scalar property (i.e., Weber's law) on a linear scale, where Bayesian integration on that linear scale induces a logarithmic compression of responses. Hence, it's simply not clear which theoretical assumption should be considered more fundamental. At least it's not a theoretical issue our model (or empirical data) can solve - and as such, we have decided to stay away from this discussion.

A second issue is whether Bayesian integration happens on a logarithmic or linear scale. The MLN model assumes that, at some point the gaussian likelihood is transformed to a log-normal, i.e. a logarithmic encoding of time. However, the MLN additionally assumes that Bayesian integration happens on a linear scale. Therefore, the crux of the difference between 'pure logarithmic' Bayesian models and MLN is whether Bayesian integration happens on an early logarithmic scale, or a late linear scale. While we are unable to arbitrate between these possibilities, some empirical evidence (Remington et al., 2018) suggests that, in temporal reproduction paradigms, Bayesian integration may take place at late rather than early stages (although this wasn't explicitly tested in the context of log vs linear scales), even though the work of Damsma, Schlichting, and Van Rijn (2021) suggests that the prior does influence early perceptual stages.

We now introduce these considerations in the introduction, as they indeed provide important context for our work.

2. p4. the final paragraph. I do not fully understand the logic here. My understanding is that if you model the prior with an internal logarithmic scale, you might have a simpler Gaussian prior with the scalar property and the skewed central tendency (biased more with the long relative to the short duration).

We agree with the reviewer that Bayesian integration is simpler and more straightforward with Gaussians than with log-normals (although simply transforming from logarithmic to linear scales is all that's needed). Nevertheless, this benefit is not apparent anymore when we're using a mixture distribution for the prior, which is a central tenet of our MLN model.

3. Model comparisons based on tp only

pp. 8-9 Fitting the student sample: Here three models were compared based on the reproduction (tp), but the differences in terms of RMSEs are relative small (0.032, 0.028 and 0.027, respectively). I wonder if the prediction from MLN is better than others for accounting both tp and the standard deviations of tp (i.e., predicting the distribution of tp). Or, in other words, would the MLN predicts better for the observed tp and the scalar property? Same comment can be applied for the model comparison of MCI data.

Indeed, the average difference in model fit in terms of RMSE is small, but it should be noted that these are averages per quantile, per ts, and per individual. Additionally, we believe that the distribution of responses for every tp is accounted for by the modeling, because we fit the models to the response quantiles, not just the mean.

We also assessed the fit of each model to the scalar property (or in this case, the lack thereof). Here, we plot the Coefficient of Quartile Variation (CQV) which is an appropriate measure of CV for skewed data:

$$CQV = (Q3 - Q1) / (Q3 + Q1)$$

The empirical data show a downward sloping CQV (see, e.g., Ren et al. 2021). As can be seen, the models capture the CQV averaged over ts durations quite well, but only the Gaussian and MLN model qualitatively capture the general trend of the data. Still, neither model seems to fully capture the magnitude of this negative trend, suggesting that an additional 'time-independent' source of variance may be needed (Ren et al., 2021).

Minors

4. p4. line 51. "... the locations of the other components reflect ..." is not clear to me. "what other components do you refer to?"

Here we referred to the mixture distribution components. We now specify this where relevant.

5. p4. line 55-56. "We will assume that the means of the distributions will be distributed as a geometric series, resembling the type of distribution of the presented stimulus durations". It is not clear to me. Were your durations sampled in geometric spacing or are there any literature supporting this assumption?

The durations presented were geometrically spaced, and we assumed out of parsimony that the means of the distributions follow the same spacing. We have changed the wording, hopefully preventing this confusion.

6. p.6. Analysis code of Maass and Van Rijn (2018) is available, but the data are missing in the OSF repository. uppsi

Thank you for spotting this omission, this is now resolved.

7. p 12. line 51. "there is no evidence for a difference for w_p between both groups" This needs statistical support.

Bayes Factors were plotted in Figure 7; we have now also added them to the text, including a cautionary sentence regarding the magnitude of the effect.

Reference:

Ren, Y., Allenmark, F., Müller, H. J., & Shi, Z. (2021). Variation in the “coefficient of variation”: Rethinking the violation of the scalar property in time-duration judgments. *Acta Psychologica*, 214, 103263.